# Reconstructing the evolution history of networked complex systems

Junya Wang[1,11], Yi-Jiao Zhang [2,11], Cong Xu[2,11], Jiaze Li[3], Jiachen Sun[4], Jiarong Xie [5,6], Ling Feng [7,8], Tianshou Zhou [9] & Yanqing Hu [2,10] ✉

The evolution processes of complex systems carry key information in the systems' functional properties. Applying machine learning algorithms, we demonstrate that the historical formation process of various networked complex systems can be extracted, including protein-protein interaction, ecology, and social network systems. The recovered evolution process has demonstrations of immense scientific values, such as interpreting the evolution of protein-protein interaction network, facilitating structure prediction, and particularly revealing the key co-evolution features of network structures such as preferential attachment, community structure, local clustering, degree-degree correlation that could not be explained collectively by previous theories. Intriguingly, we discover that for large networks, if the performance of the machine learning model is slightly better than a random guess on the pairwise order of links, reliable restoration of the overall network formation process can be achieved. This suggests that evolution history restoration is generally highly feasible on empirical networks.

As generic representations of vastly different complex systems, complex networks[1–3] are widely used in different areas across biology[4–7], ecology[8,9], social science[10,11], etc. Networks represent the internal interactions between the various components within the systems. For these networked complex systems, evolution is their most striking feature. However, what distinct underlying mechanisms do they follow when evolving from simple structures to the current complex forms? How do patterns and functionalities emerge during the evolutionary process of the networks? What are the future directions of evolution? These are key scientific questions about complex systems that have challenged the academic community for a long time.

The structures of most complex networks from biology, ecology, and human society are very complicated. Characteristics such as

hierarchical community structure[2], (dis)assortativity[12], local clustering[13], motifs[14], etc., are ubiquitous in complex networks. These make it challenging to comprehensively capture the evolution mechanisms that generated such complex structures with concise rules, as existing researches on the evolution of complex networks typically only focus on certain specific features of real-world networks. For instance, the well-known preferential attachment (PA) mechanism[15] can only explain the scale-free property of a network's degree distribution but not other features, and sometimes even lead to contradictions with other features (e.g., networks generated by PA have zero local clustering coefficient[16] and no communities[17]).

In this work, by employing graph neural network (GNN) models, we demonstrate that the evolution process of a network can be

[1]School of Systems Science and Engineering, Sun Yat-sen University, Guangzhou 510006, China. [2]Department of Statistics and Data Science, College of Science, Southern University of Science and Technology, Shenzhen 518055, China. [3]Department of Data Analytics and Digitalisation, School of Business and Economics, Maastricht University, Maastricht 6200MD, The Netherlands. [4]Tencent Inc., Shenzhen 518000, China. [5]Center for Computational Communication Research, Beijing Normal University, Zhuhai 519087, China. [6]School of Journalism and Communication, Beijing Normal University, 100875 Beijing, China. [7]Institute of High Performance Computing (IHPC), Agency for Science, Technology and Research (A*STAR), Singapore 138632, Singapore. [8]Department of Physics, National University of Singapore, Singapore 117551, Singapore. [9]School of Mathematics, Sun Yat-sen University, Guangzhou 510275, China. [10]Center for Complex Flows and Soft Matter Research, Southern University of Science and Technology, Shenzhen 518055, China. [11]These authors contributed equally: Junya Wang, Yi-Jiao Zhang, Cong Xu. ✉e-mail: yanqing.hu.sc@qq.com

reconstructed with high precision. Validated computationally, our developed theory indicates that such reconstruction can be done reliably even with slightly better than a random guess on the pairwise temporal order of links. The recovered evolution trajectories enable us to discover concise rules in the complex evolution process of networks, and capture the emerging process of key characteristics of a network that previous theories were unable to capture collectively with concise rules (e.g., community structure, local clustering, (dis) assortativity, etc.). In addition, we show that such high-resolution evolution trajectories have important practical applications in facilitating network structure prediction, interpreting the evolution of protein-protein interaction networks, as well as revealing the co-evolution mechanisms of preferential attachment and community structure.

## Results

### Methodological framework

The main purpose of this study is to restore the growing edge sequence for an evolving network based on its final structure (see Fig. 1a). We achieve this goal in two steps using machine learning techniques (illustrated in Fig. 1b, c). First, for networks where a small fraction of the edge generation sequence is available, we build a supervised machine learning model leveraging the network topology and known history to infer the formation process of the entire network. Second, for networks where only the final structure is available, we adopt a transfer learning approach[18] in which a machine learning model trained on a similar network as described in the first step is applied to the target network.

In the first step, the edges in the final structure of a network with partial evolution history (Network *A*) are embedded into a low-dimensional space[19–23]. Then the edges are paired and used to train the

machine learning model for predicting the relative generation order of any two edges in the network. Note that the training set includes only edge pairs with known generation order which can be directly obtained from the evolution history. The concrete model proposed here is an ensemble model consisting of six comparative paradigm neural network (CPNN) models[24] and a classical edge feature (see Supplementary Fig. S1). Once the predicted generation order of any two edges is obtained, a ranking algorithm, the Borda's method[25], is applied to find an ordered sequence of all edges so that the formation process of the full network can be recovered accordingly. More details of our approach with regard to embedding the edges, building the ensemble model, and implementing the ranking algorithm are provided in the Methods section and the Supplementary Sections 1 and 2. In cases where the purpose is to restore high temporal resolution of a network's historical evolution process from very low temporal resolution data like the case in some of the biological/ecological networks, this first step is enough for the purpose. In certain applications where even the low resolution of a network's evolution history is unknown, we then proceed to the next step. In the second step, the edges in a network of the same domain without any historical information (Network *B*) are embedded into a low-dimensional space and aligned with that of Network *A* through a linear transformation, which is the key to successful transfer learning (details in Methods section). Lastly, the ensemble model trained on Network *A* as well as the same ranking algorithm in the first step can be used to infer the formation history of Network *B*.

### Restoration of network evolution trajectory

To demonstrate the effectiveness of the proposed method, we apply it to 17 real-world networks with multiple snapshots of their evolving processes. The 17 networks include five protein-protein interaction (PPI) networks[26–28], a world trade web[29,30], six collaboration networks[31,32], two

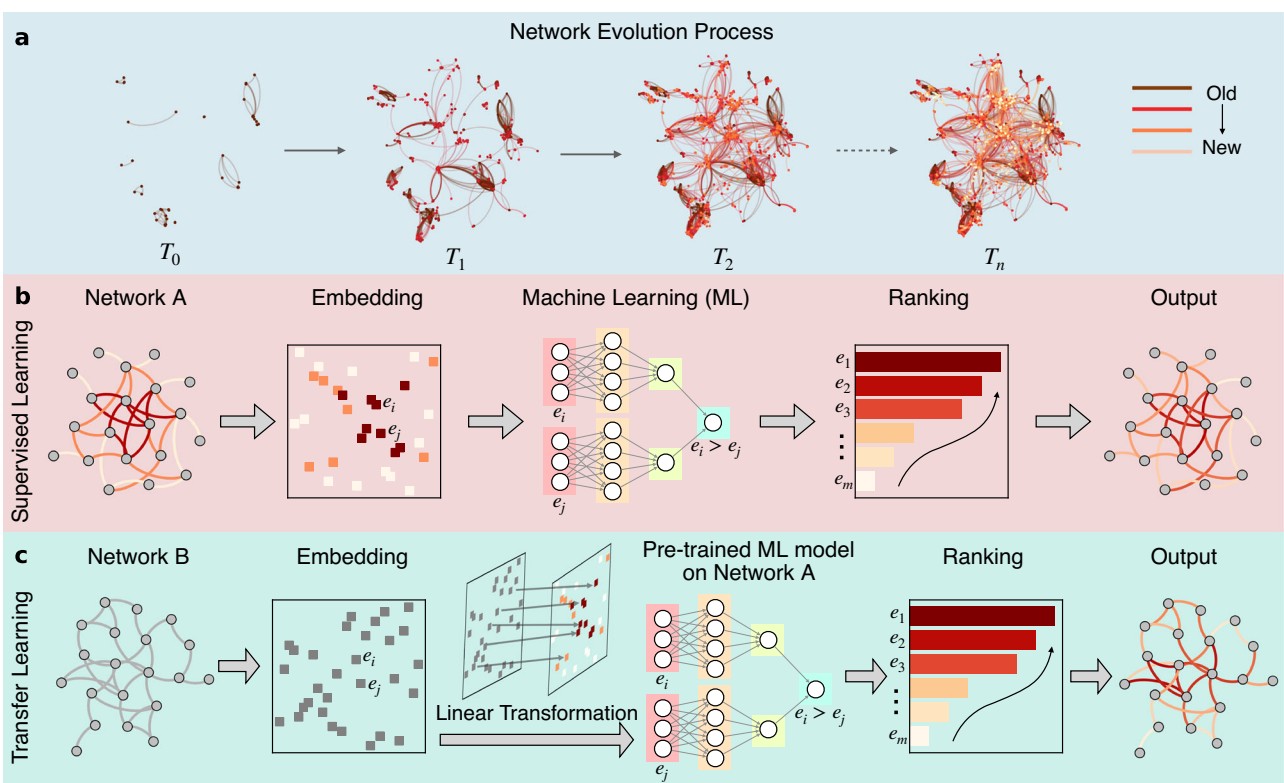

**Fig. 1 | The network formation process and its restoration. a** Illustration of a network formation process. At each snapshot $T_0$, $T_1$, $T_2$, ..., $T_n$, some new edges are added (darker edges appeared earlier). The goal of this study is to restore the generation order of the edges based on the final network structure at $T_n$.

**b**, **c** Diagram of the proposed approach to restoring the temporal sequence of edges for a network with partial evolution history or without any historical information.

animal interaction networks[33,34], and three transportation networks[35] (full listing is given in Supplementary Section 3). The generation time of an edge is assigned as the timestamp of the snapshot in which it appears for the first time. In this way, the relative generation order of edges from different snapshots can be obtained. Depending on the granularity of the snapshots, different networks have varying amounts of edge pairs with distinguishable generation orders.

The performance of the proposed approach in restoring the network formation process with partial history is quite satisfactory (the comparison between restored, random, and real evolution trajectories for some networks can be seen in Supplementary Section 4 and the Supplementary Movie 1–3). First of all, it is surprising that we only need a small percentage of edge pairs to train the ensemble model to obtain high accuracy. We define $x$ to be the number of edge pairs with correctly predicted generation order divided by the total number of edge pairs in the test set. As Fig. 2a shows, the pairwise accuracy $x$ of the ensemble model in predicting the relative generation order of any two edges increases rapidly to over 75% as more edge pairs are used to train the model and saturates when the percentage reaches just 5% (more results can be found in Supplementary Section 5 and Supplementary Fig. S8). While $x$ represents the accuracy of the intermediate results of our approach, we are also interested in quantifying the error of the final output, i.e., the restored temporal sequence of edges. Let $\mathcal{E}$ denote the overall error of the restored edge sequence, we would like to further explore how is $\mathcal{E}$ related to $x$. Denote $\alpha_i$ as the position of edge $i$ in the true edge sequence (e.g., $\alpha_i = i$, larger $\alpha_i$ means that edge $i$ joined the network later) and $\widehat{\alpha}_i$ as its corresponding position in the output sequence of our approach. Then $\boldsymbol{\alpha} = (\alpha_1, \alpha_2, ..., \alpha_E)$ and $\widehat{\boldsymbol{\alpha}} = (\widehat{\alpha}_1, \widehat{\alpha}_2, ..., \widehat{\alpha}_E)$ are the ground-truth sequence and the restored sequence, respectively. Thus, $D_i = \alpha_i - \widehat{\alpha}_i$ measures the error of edge $i$ so that the overall error $\mathcal{E}$ of the entire sequence can be defined as the root-mean-squared error (RMSE) normalized by $E$, i.e.,

$$\mathcal{E} = \sqrt{\frac{1}{E} \sum_{i=1}^{E} \left( \frac{D_i}{E} \right)^2}. \tag{1}$$

This definition of the overall error is theoretically equivalent with other measures for assessing the correlation between two ordered sequences, including the Kendall's $\tau$[36] and Spearman's $\rho$[37] (please refer to the Supplementary Section 6 for more details). We choose the $\mathcal{E}$ in Eq. (1) as the measure of performance because its physical meaning is more intuitive compared to the other measures. After mathematical derivation (see details in the Methods section and Supplementary Section 6), the theoretical relationship between $\mathcal{E}$ and $x$ is

$$\mathcal{E}^{\text{theory}} = \frac{\sqrt{x(1-x)}}{2x-1} \frac{1}{\sqrt{E}}, \tag{2}$$

where $x \gg 0.5 + \frac{1}{4\sqrt{E}}$.

Equation (2) shows that the overall error of the restored edge sequence is inversely proportional to the square root of the number of edges, suggesting that our approach has a huge advantage for networks with a rich number of edges. In other words, when the number of edges is large enough, we only need a machine learning model with accuracy slightly better than a random guess for predicting the relative generation order of any two edges to make the overall error small. This is a really nice property and consistent with the results shown in Fig. 2b.

Investigating further on this point, the distributions of $D_i/E$ (see Fig. 2c) are bell-shaped and symmetric about zero with the spread of the distribution determined by $E$ and $x$, i.e., the spread decreases with $E$ and $x$. While these results are based on simulations with fine-grained ground-truth sequence, the one we have in practice is typically coarse-grained. Therefore, we also draw the distributions of $D_i/E$ with coarse-grained ground-truth sequence for real-world networks (see Fig. 2d, e). The results demonstrate that the distributions of $D_i/E$ based on real

data are in accordance with those obtained by simulations which reflect the theoretical results. For details on drawing Fig. 2c–e, please refer to the diagram shown in Fig. 2f and the pseudo code in the Methods section. However, it is worth noting that for real-world networks that lack fine-grained ground truth, it is a challenging problem to verify the credibility of the restored network evolution trajectory. This is a generic problem for many machine learning techniques. A preliminary discussion on this topic is provided in the Supplementary Section 7.

## Transfer learning
Finally, the performance of our transfer learning approach in restoring the network formation process for networks without any historical information is explored. We compare the performance of transfer learning (i.e., aligning the vector representations of Network $B$ with that of Network $A$, see details in the Methods section and the Supplementary Section 8) with that of direct validation (the vector representations of Network $B$ are fed directly into the ensemble model trained on Network $A$). The results on different synthetic network models[15,38,39] are summarized in Table 1. We can see that the accuracy of transfer learning is much higher than that of direct validation, indicating that our approach is able to restore network formation processes well with only the final structure.

## Interpretation of the evolution of PPI network
Having been able to reliably reconstruct the evolution history of networked systems, we can carry out rich scientific investigations based on the reconstructed edge sequence, ranging from understanding the evolution or emergence of functional properties, extracting fundamental evolution mechanisms, to even facilitating practical problems like structural predictions for future evolution of complex networks. Here we show that our restored edge sequence can help understand the evolutionary process of living systems. The evolution trajectory of PPI networks is critical for understanding the fundamental mechanisms of cellular processes and the emergence of complexity of life forms. This enables researchers to gain insights into the function of protein organization[40], the development of new biological functions[41], and the selection mechanisms driving network evolution[42]. However, to the best of our knowledge, there is no complete data on the evolution trajectory of PPI so far due to the lack of paleontological data. This is where our network restoration method comes into play.

By applying our method to PPI networks, we find that proteins with specific functions appear in an order reflecting the evolutionary patterns of life. Take the PPI network for fungi as an example, Fig. 3a shows the restored network snapshot corresponding to the immemorial times which exhibits some distinct cluster structures. Interestingly, we find that nearly every cluster is composed of proteins with consistent functionality. According to the order of the edges, we count the absolute number of proteins by function over time and calculate the proportion of proteins with different functions added (see Fig. 3b, c). These results suggest that the evolution of the PPI network focused early on basic functions at the molecular level like protein synthesis and gene expression regulation (e.g., [J] translation, ribosomal structure, and biogenesis), then shifted to the maintenance of genetic information, and eventually towards advanced functions at the cellular level like cell division and inheritance of genetic material (e.g., [D] cell cycle control, cell division, chromosome partitioning). Note that we arrive at it based solely on the limited historical information of the PPI network without referring to much biological knowledge. We believe that our work provides biologists with a novel way to explore more principles underlying the evolution of complex life.

## Revealing the evolution mechanisms
We then show that our restored edge generation sequence not only enables us to reproduce the growth mechanisms with the same

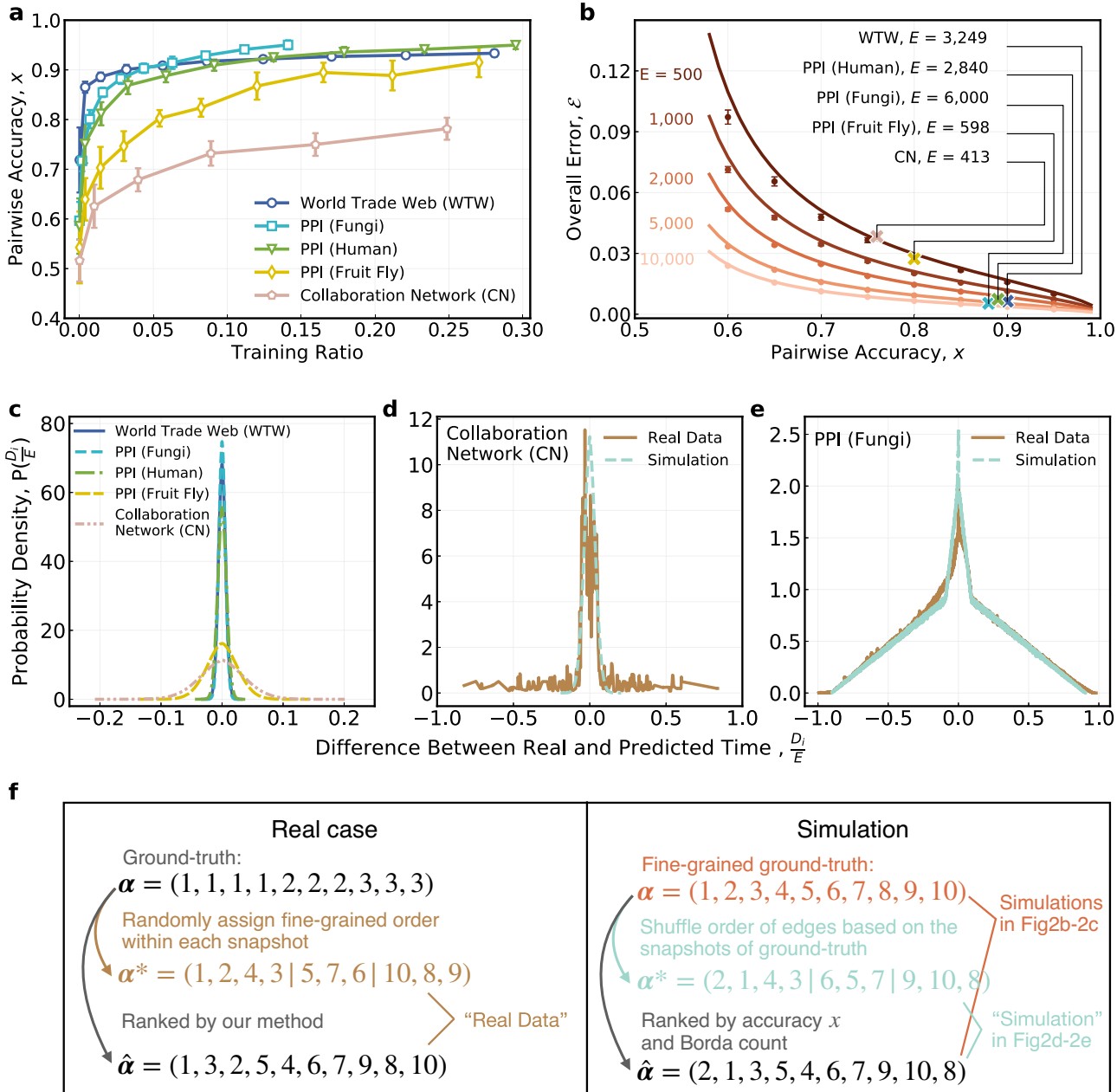

**Fig. 2 | Performance of the ensemble model and the restored edge sequence.**
**a** Test accuracy of the ensemble model as a function of the percentage of edge pairs used for training. Each data point with error bars marks the corresponding simulation results (average ± standard deviation of 100 simulations), the same for **b**. **b** Overall error $\mathcal{E}$ as a function of the accuracy $x$ of the ensemble model for different numbers of edges $E$. The solid curves represent the theoretical results from Eq. (2) and the colored crosses stand for the simulation results using the $E$ and $x$ of five real-world networks. **c** Simulated distributions of $D_i/E$ using the $E$ and $x$ of five real-world networks. Specifically, assuming the ground-truth sequence $\alpha = (1, 2, ..., E)$, $100(1 − x)\%$ of all edge pairs are randomly selected and artificially assigned the wrong generation order while the remaining edge pairs are assigned the correct one. Then, the restored edge sequence $\hat{\alpha}$ is obtained by applying the ranking algorithm on the artificially predicted order of all edge pairs and $D_i$'s are calculated

accordingly. **d**, **e** Comparisons between the real and simulated distributions of $D_i/E$ based on the collaboration network (CN) and the PPI network (Fungi). **f** Diagram illustrating how the distributions in **c**–**e** are obtained. The left and right panels show the calculation of $D_i$ under a real case when we only know the coarse-grained ground-truth sequence and a simulation when we know the fine-grained ground-truth sequence, respectively. For the real case, $D_i$ cannot be calculated directly as $\alpha_i − \hat{\alpha}_i$ so the idea is to consider an intermediate sequence $\alpha^*$ by randomly assigning fine-grained order to edges added within the same snapshot and $D_i$ is calculated as $\alpha_i^* − \hat{\alpha}_i$ instead. Then the distribution of $D_i/E$ is obtained by averaging over 5000 $\alpha^*$'s to take the randomness into account. For the simulation, the calculation of $D_i$ follows a similar procedure to match with the real case. The results under the real case and simulation are labeled as "Real Data" and "Simulation" in **d** and **e**. See Algorithms 2-3 in the Methods section for more details.

preferential strength as the original network, but also to observe richer mechanisms related to community structures in the network other than preferential attachment (PA). PA is a commonly-known mechanism in the growth of real-world networks, producing networks with power-law degree distributions[15,43,44]. To date, researches on the growth mechanisms of networks with power-law degree

distributions have been largely confined to PA or its variants while deeper, especially those about sub-network functions, remain an under-explored area.

From Fig. 4a−c, it is observed that the restored growth process of many real-world networks shows the PA phenomenon with the same strength as the original network, demonstrating that our method can

**Table 1 | The accuracy of transfer learning and direct validation**

| Train Net. | Fitness (N = 500) | | Fitness (N = 1000) | |
|---|---|---|---|---|
| Test Net. | $x^T$ | $x^D$ | $x^T$ | $x^D$ |
| BA (N = 500) | **0.853 ± 0.004** | 0.694 ± 0.03 | **0.859 ± 0.001** | 0.697 ± 0.012 |
| BA (N = 1000) | **0.832 ± 0.003** | 0.685 ± 0.020 | **0.839 ± 0.001** | 0.679 ± 0.013 |
| PSO (N = 500) | **0.830 ± 0.007** | 0.682 ± 0.025 | **0.845 ± 0.002** | 0.653 ± 0.002 |
| PSO (N = 1000) | **0.836 ± 0.007** | 0.701 ± 0.019 | **0.848 ± 0.001** | 0.664 ± 0.019 |

"Train Net." and "Test Net." refer to the networks used to train and test the ensemble models, respectively. The pairwise accuracy of the ensemble model evaluated on "Test Net." under transfer learning is denoted as $x^T$ and that under direct validation is denoted as $x^D$. The network models used are the Barabási–Albert (BA) model[15], the popularity-similarity-optimization (PSO) model[38,53], and the fitness model[39] (detailed information can be found in Supplementary Section 8). The value in the table represents the average accuracy and its standard deviation from 10 simulations. Better results are highlighted in boldface. Swapping the training and testing network models yields consistent results.

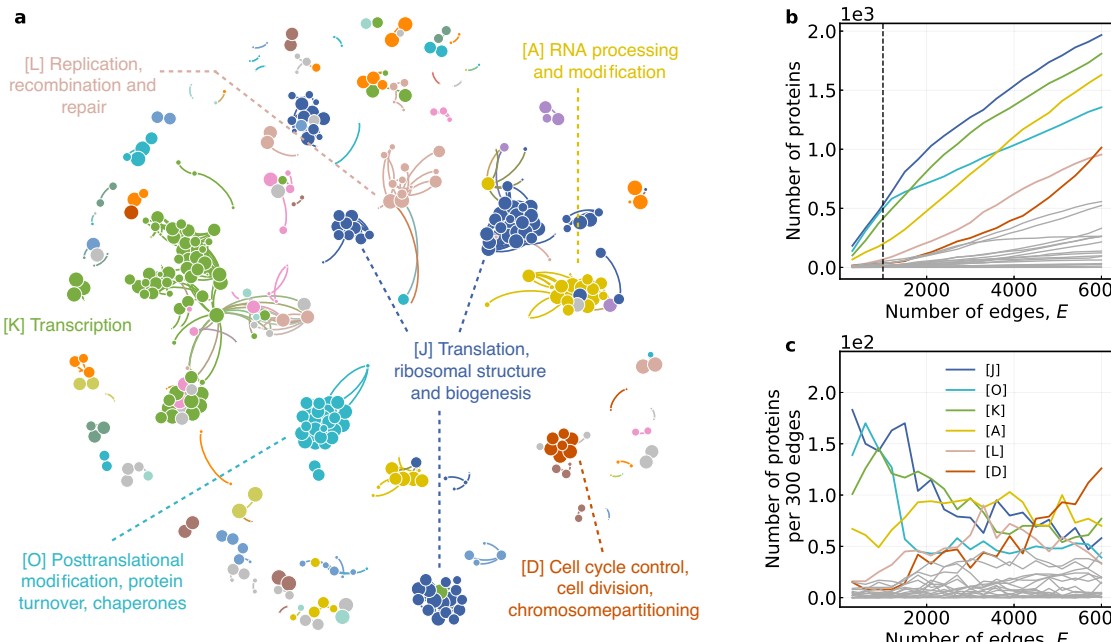

**Fig. 3 | Application on the PPI network for fungi. a** The restored network structure and protein functional clusters at the time that the first 1000 edges were added. The size of the nodes indicates the order in which the nodes appear, the nodes added first (i.e., the nodes corresponding to the edges that are added first) are larger. The colors of the nodes represent different functions of the proteins (full protein functions are listed in Supplementary Table S8). It can be seen that in the evolution process of the PPI network, interactions between proteins form protein clusters with specific functions. **b** The number of proteins by function over time counted according to the order of the edges. Proteins at both ends of each edge are considered. **c** The number of proteins with different functions added in each interval of 300 edges. The functions represented by each capital letter can be found in **a**.

well capture the growth mechanism of the networks and the results are highly reliable (the detailed information can be found in Supplementary Section 9). Then taking the PPI network for fungi as an example, we investigate the meso-level evolution process of real-world PPI networks on the basis of our restored results. Concretely, a meso-level protein network is constructed with each node being a protein functional community, i.e., a collection of proteins with the same function. Edges between proteins with the same function in the original PPI network are reflected as self-connected edges in the new network. The upper row of Fig. 4d displays the evolution process represented by the adjacency matrices of the meso-level protein network based on our restored edge sequence while the lower row provides those obtained when new edges are added purely according to the PA rule. By comparison, we find that the growth process of our restored network is significantly different from that under the pure PA mechanism. Specifically, the restored results show that newly added edges tend to connect proteins within the same community, allowing the PPI network to maintain a strengthened community structure during growth. On the contrary, newly added edges tend to connect proteins between communities under the pure PA mechanism, weakening the existing community structure in the evolution process. The adjacency matrix and protein function network based

on the real network is displayed in Fig. 4h, i, showing that our restored network highly agrees with the real network. Figure 4g further demonstrates that our restoring method captures the strong community structure in the real network while the pure PA rule fails to achieve. The restored co-evolution of community structure and preferential attachment provides commendable data support for understanding the relationship between the structure and function of networks.

Moreover, we also study how likely that nodes with similar degree tend to be connected (i.e., degree-degree correlations[12]) and how clustered the connections are (i.e., local clustering[13]). Figure 5 displays the results for selected real-world networks and the full results can be found in Supplementary Section 9. The big gap among the results based on our restored edge sequences, the random edge sequences, and the edge sequences assuming pure PA rule along with the high concordance between the restored and the real edge sequences demonstrate that rich characteristics of a network can be recovered based on the restored evolution process.

**Facilitating structure prediction**

Structure or link prediction is a task that aims to predict new edges based on existing ones in a network, which is widely used in

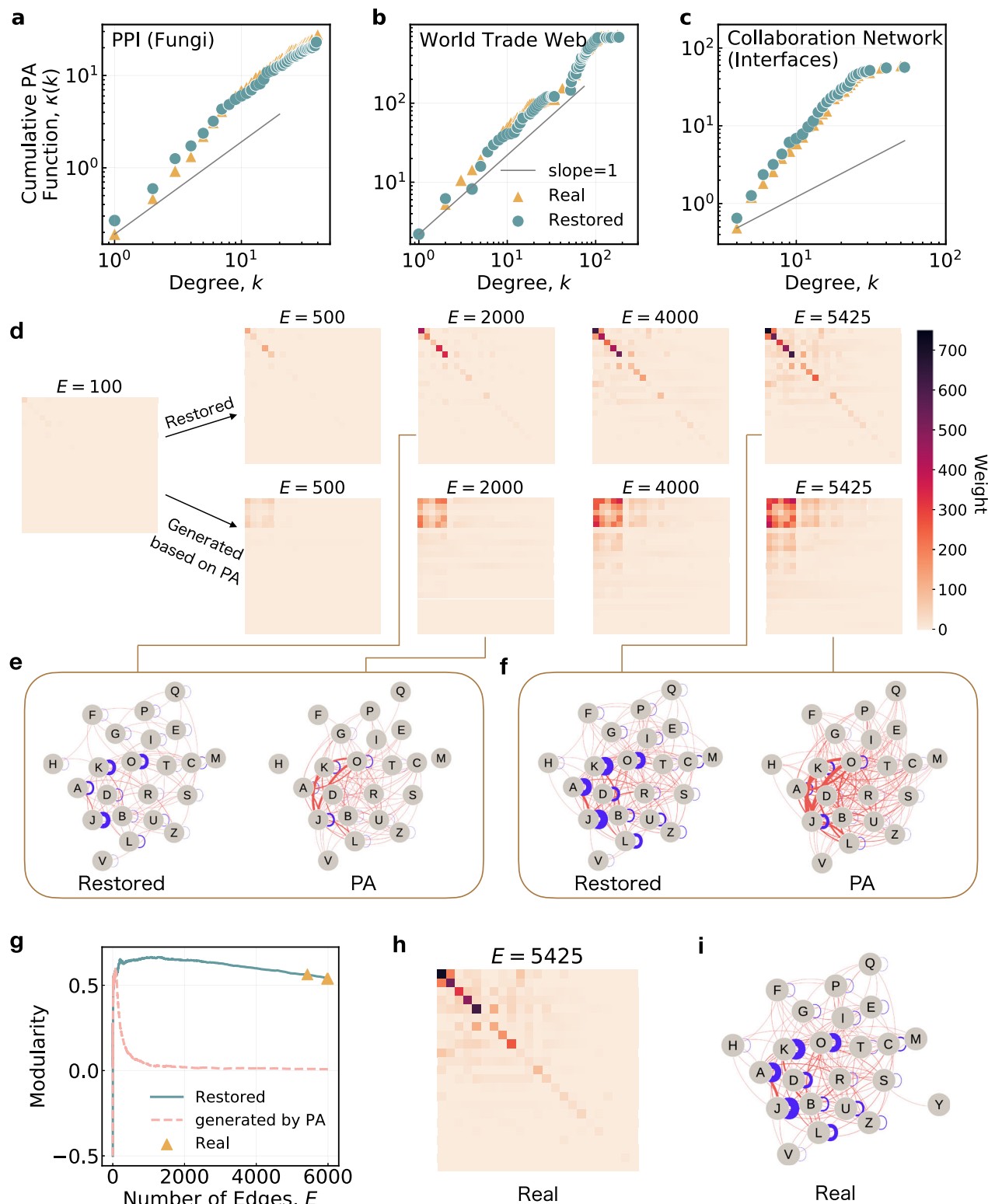

drug development[45–47], protein interaction prediction[4,6] and recommendation systems[48,49]. Here we show that the edge generation order produced by our method can be used in link prediction and improve the prediction accuracy significantly. Specifically, we regard the network whose edge generation sequence has been restored by our approach as a tensor, denoted by $\mathbf{Z}$ with elements $\mathbf{Z}(i_1, i_2, \widehat{\alpha}_i)$ which takes value 1 if edge $i$ is generated at position $\widehat{\alpha}_i$ and 0 otherwise ($i_1$, $i_2$ are the two nodes of edge $i$), and employ the collapsed weighted tensor method[50] to define a weighted adjacency matrix $\mathbf{X}$ with entries $\mathbf{X}(i_1, i_2) = \mathbf{Z}(i_1, i_2, \widehat{\alpha}_i) \times \theta^{\max(\widehat{\alpha}) - \widehat{\alpha}_i}$ $(\theta \in (0, 1))$. Then by applying the truncated singular value decomposition algorithm (TSVD)[50] on $\mathbf{X}$, the predicted scores for all candidate edges to be added in the future can be obtained. The candidates with larger predicted scores are more likely to be added in the future. Figure 6 clearly demonstrates that the restored edge generation order can significantly improve the link prediction performance up to several times for some networks. Significant improvements can also be found by implementing other classical link prediction algorithms

**Fig. 4 | Underlying growth mechanism in the restored network evolution processes.** Cumulative PA function $\kappa(k)$ for **a** PPI network (Fungi), **b** World Trade Web, and **c** Collaboration network (Interfaces). In each figure, the yellow circles and blue triangles are the results of the ground-truth and restored evolution processes, respectively. If the growth of a network follows the PA rule, the rate at which a node with degree $k$ acquires new edges should be positively correlated with $k$ and the cumulative PA function $\kappa(k)$ is expected to grow superlinearly (see Supplementary Section 9 for details). So, the solid gray line with slope = 1 represents the case in which PA is absent. **d** Adjacency matrices of the evolution process for the protein function network generated by the PPI network (Fungi). Proteins with the same function in the network are treated as a single node to form a simplified protein function network where the edges represent the interactions between proteins with weights being the number of protein interactions. The upper row shows the results based on our restored temporal edge sequence while the lower row shows

those based on a simulation study assuming the pure PA rule. The simulation is performed by adding edges according to the PA rule and keeping the average node degree consistent with the real network (details are provided in Supplementary Section 9). **e**, **f** Visualizations of the protein function network in **d** when the number of edges are $E = 2000$ and $E = 5425$. Letters marking the nodes denote the protein functions (with specific meanings listed in Supplementary Table S8), and the self-connected and non-self-connected edges are respectively displayed in blue and red. **g** The modularity[54] of the PPI network (Fungi). The yellow triangles represent results computed at the real snapshots of the networks. The blue solid lines and pink dashed lines are results based on edge generation order by our reconstruction method and the pure PA rule, respectively. **h**, **i** Adjacency matrix and protein function network of the PPI network (Fungi) obtained at the first real snapshot (i.e., $E = 5425$).

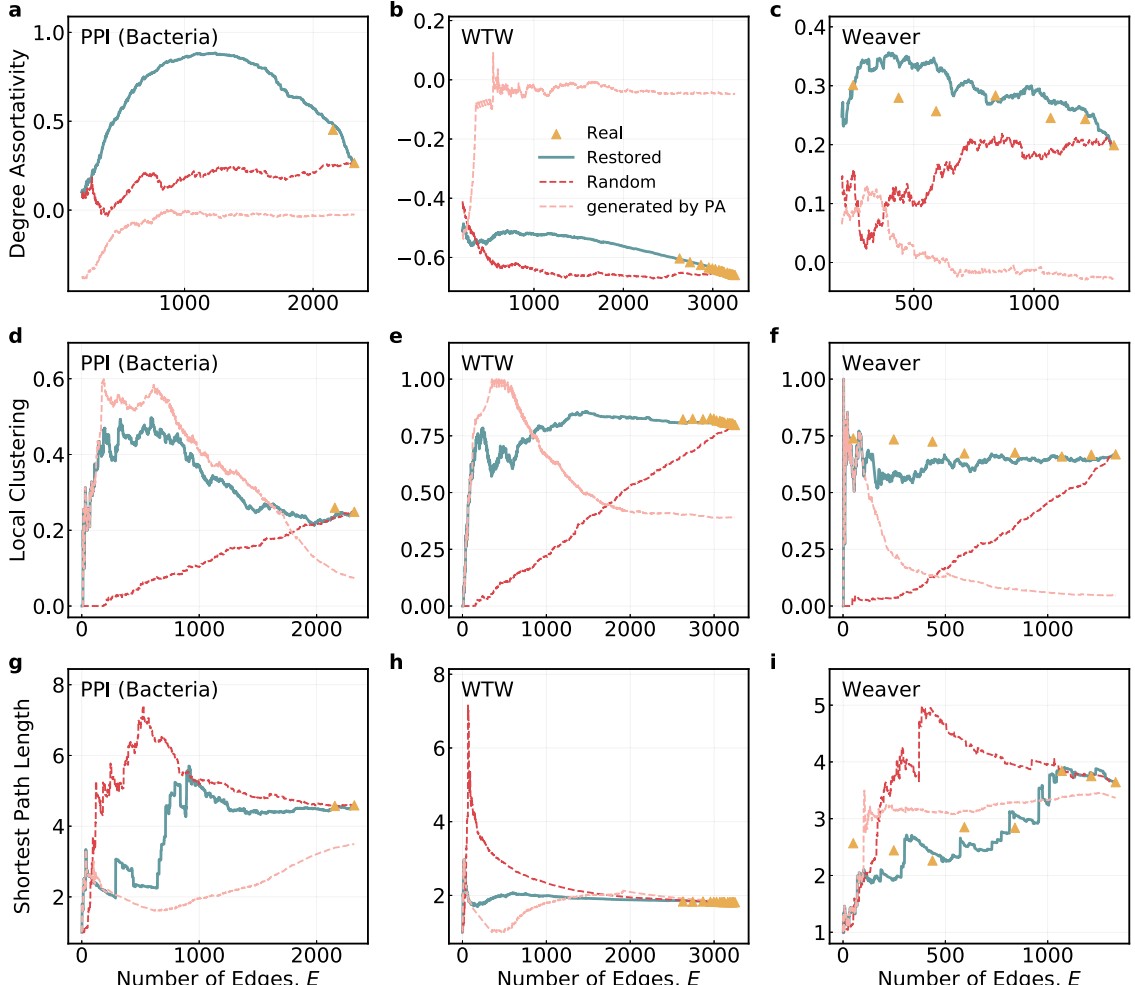

**Fig. 5 | Assortativity coefficient, local clustering coefficient, and shortest path length for the restored evolution processes.** The assortativity coefficient for **a** PPI network (Bacteria), **b** World Trade Web (WTW), and **c** Animal network (Weaver). The average local clustering coefficient for **d** PPI network (Bacteria), **e** WTW, and **f** Animal network (Weaver). The average shortest path length for **g** PPI network (Bacteria), **h** World Trade Web (WTW), and **i** Animal network (Weaver). The yellow triangles represent results computed at the real snapshots of the networks. The

blue solid lines and red dashed lines are results based on edge generation order by our restoring method and by random assignment, respectively. The pink dashed lines are results for networks generated assuming the pure PA rule. Note that due to the presence of disconnected components during the evolution process of a network, the computation of the average shortest path length only involves pairs of nodes that can be connected.

besides TSVD[51,52] on our restored edge sequence (see more results in Supplementary Section 10). It is noteworthy that after a decade of development, the design of link prediction algorithms has hit a roadblock and it is not easy to achieve such a significant performance boost.

## Discussion

The problem of restoring the system structure is of great importance in many fields. In this article, we address the fundamental problem of restoring the structure evolution trajectory of networked complex systems, and demonstrate that the problem can be resolved with high

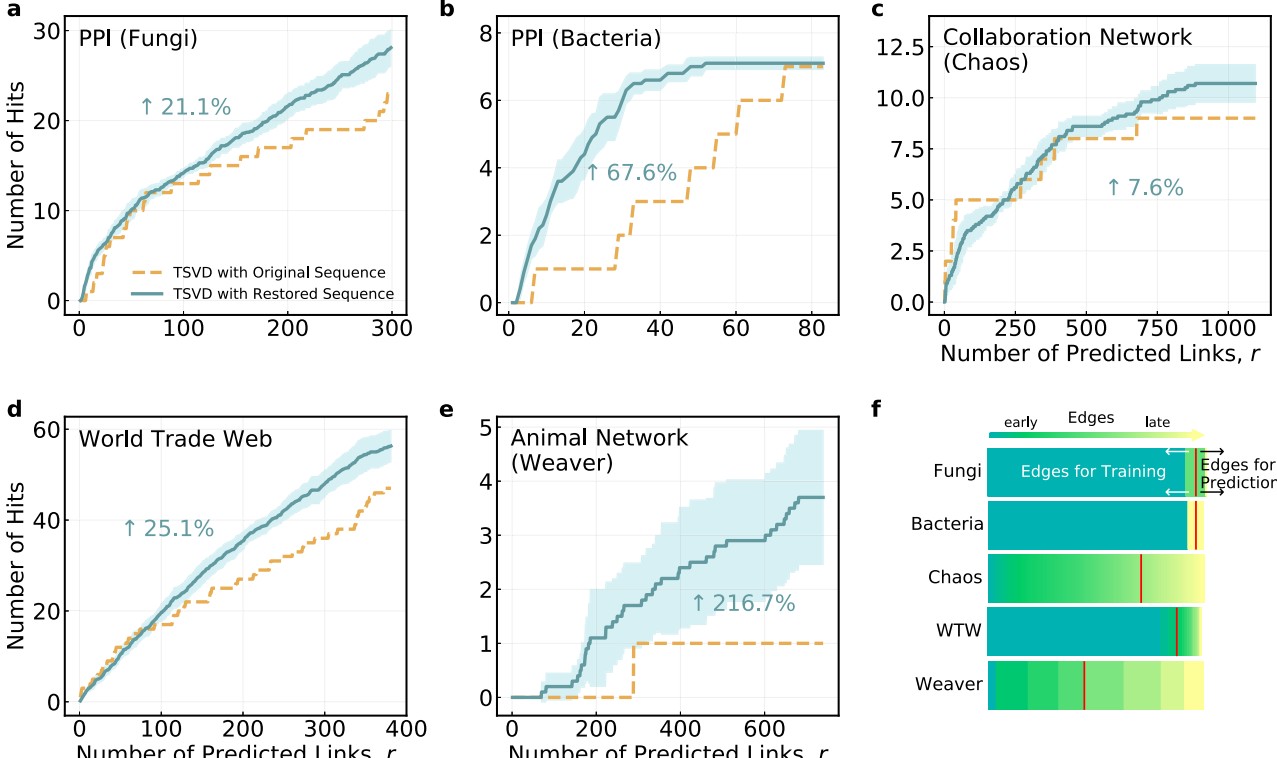

**Fig. 6 | Performance of link prediction.** Number of hits (correctly predicted edges) obtained by using the original (yellow dashed lines) and restored (blue solid lines) edge generation order on **a** PPI network (Fungi), **b** PPI network (Bacteria), **c** Collaboration network (Chaos), **d** World Trade Web, **e** Animal network (Weaver). For each network, we first remove the edges added in the last few snapshots. The number of edges removed is selected according to the number of snapshots in real network data (see **f** and Supplementary Table S10 for more details). Let $\mathbf{X}^{original}$ and $\mathbf{X}$ be the collapsed weighted tensor constructed based on the original (i.e., the real network with a few snapshots) and the restored edge generation order, respectively. Then $\mathbf{X}^{original}$ and $\mathbf{X}$ are used to calculate the corresponding score matrices and obtain the predicted future edges. The number of edges removed in the real network data that appear in the first $r$ predicted future edges is termed the "Number of hits" and used to evaluate the link prediction performance. The percentage of improvement is computed based on the area under the curve. The weight parameter and the truncation parameter in the link prediction algorithm are tuned to get the optimal results for both the original and restored sequences, respectively. Results based on the restored sequences are averaged over 10 repeated simulations, with the light blue areas representing the 95% confidence intervals. **f** An illustration of the edges removed in different networks. Edges are arranged by real generation order with those added earlier to each network displayed in darker color. Edges after the red line are removed as the test set of link prediction. Note that the restored sequences used for link prediction are obtained using the same method as other tasks in this work but on networks without edges for prediction.

accuracy based on graph neural network techniques, especially for networks with a large number of edges. With the restored edge formation history, the performance of link prediction algorithms can be greatly improved and the network evolution mechanisms can be revealed.

Note that there are some limitations in this work: 1) We assume that the edges in a network will always exist after they are generated. However, in many real-world systems, many nodes and edges diverge or even disappear. 2) For many real-world networks, there may be only a small number of edge pairs with time information (i.e., edge pairs with distinguishable generation order) and the generation time of these edge pairs may be biased. For example, for the five PPI networks used in this work, there are only a few snapshots and edges with time information are mostly from the last snapshot. In this case, how to measure the credibility of the restoration results is also a problem worthy of in-depth study. 3) Our current transfer learning technique can only be successfully implemented on artificially synthesized networks with similar generation mechanisms. The application of transfer learning to real-world networks requires further exploration. Our work is just the beginning, not the end, of the researches in this field. Nevertheless, we believe that our research provides a novel path and approach for understanding the structure formation of networked complex systems, the relationship between structure and functionality, as well as the practical application of complex networks in a broad range of research fields including life science, brain science, ecology, information science, etc.

## Methods
### The embedding methods
In our approach to restore the temporal sequence of edges for an evolving network, we first obtain the low-dimensional vector representation for each edge. Two types of representation methods are implemented in this work. One is network embedding, which learns low-dimensional representations of nodes in a network based on its topology. After getting the vectors of all nodes, the vector representation of an edge is computed as the Hadamard product of the corresponding two node vectors. Specifically, five popular node embedding methods are applied, namely Node2Vec[19], DeepWalk[20], SDNE[23], LINE[21], and Struct2Vec[22]. The other one is a vector consisting of eleven classical edge features (see Supplementary Section 1 for details). With five vectors obtained by the five network embedding methods and one vector of edge features, we have six vector representations $\mathbf{e}_i^1, \mathbf{e}_i^2, \dots, \mathbf{e}_i^6$ for edge $i$.

### The ensemble model
An important step of our approach is to predict the relative generation order of any two edges with a machine learning model. In our work, an ensemble model consisting of six CPNN models is proposed, each taking the vector representations of two edges $\mathbf{e}_i^l$ and $\mathbf{e}_j^l$ ($l = 1, 2, \dots, 6$)

as input. Each CPNN model outputs a probability that edge $i$ is added to the network later than edge $j$, generating six probabilities $o_i^1, o_i^2, \ldots, o_i^6$. Moreover, we select the feature that has the highest prediction accuracy in the training set among the eleven edge features as the "best feature" and obtain an additional output $o_i^7$ based on it (e.g., $o_i^7 = 1$ if edge $i$ has a larger value on the best feature than edge $j$ and $o_i^7 = 0$ otherwise). The final output of the ensemble model is a weighted average of all seven outputs:

$$o_i^{\text{final}} = \sum_{l=1}^{7} o_i^l w_l, \qquad (3)$$

where the weights $w_1, w_2, \ldots, w_7$ satisfy $\sum_{l=1}^{7} w_l = 1$ and are determined by grid search during training.

### The ranking algorithm
In our approach, the Borda's method, a voting-based ranking algorithm, is used to find an ordered sequence of all edges based on the predicted generation order of any two edges. Specifically in our setting, the relative generation order of any two edges is considered a ranking result so that the Borda count for edge $i$ is

$$u_i = \sum_{j=1, j \neq i}^{E} u_{ij}, \qquad (4)$$

where $u_{ij} = 1$ if edge $i$ is newer than edge $j$ and $u_{ij} = 0$ otherwise. Then the temporal sequence of all edges from old to new is determined by ranking the edges by their Borda count in ascending order.

### The theoretical relationship between $\mathcal{E}$ and $x$
A brief mathematical derivation of the theoretical result about the relationship between the overall error $\mathcal{E}$ of the restored sequence and the accuracy $x$ of the ensemble model is provided here. Without loss of generality, assume that the ground truth sequence is $\boldsymbol{\alpha} = (\alpha_1, \alpha_2, \ldots, \alpha_E) = (\frac{1}{E}, \frac{2}{E}, \ldots, 1)$ (normalized by the number of edges $E$). For the $u_{ij}$ in Eq. (4), its expectation and variance are (in subsequent derivation, the counts are normalized by $E$ for convenience)

$$\mathbf{E}(u_{ij}) = \begin{cases} \frac{x}{E}, & \text{if } \alpha_i > \alpha_j \\ \frac{1-x}{E}, & \text{if } \alpha_i < \alpha_j \end{cases}, \ \mathbf{Var}(u_{ij}) = \frac{x(1-x)}{E^2}. \qquad (5)$$

Then for the Borda count $u_i$, its expectation is

$$\mathbf{E}(u_i) = \sum_{j=1, j \neq i}^{E} \mathbf{E}(u_{ij}) = (i-1)\frac{x}{E} + (E-i)\frac{1-x}{E}$$
$$= \frac{2x-1}{E} i + 1 - \frac{E+1}{E} x \approx \frac{2x-1}{E} i + 1 - x. \qquad (6)$$

The approximation is obtained since $(E+1)/E \approx 1$ for large $E$. Treating $E$ and $x$ as constants, $\mathbf{E}(u_i)$ is a linear function of $i$ with two boundaries $\mathbf{E}(u_1) = 1 - x$ and $\mathbf{E}(u_E) = x$. In other words, the normalized Borda counts of the $E$ edges are evenly distributed over the interval $[1-x, x]$. According to the mean field theory, the position $\hat{\alpha}_i$ of edge $i$ in the restored sequence $\hat{\boldsymbol{\alpha}}$ should be the length from $1-x$ to $u_i$ divided by the total length of the interval $2x-1$, i.e.,

$$\hat{\alpha}_i = \frac{u_i - (1-x)}{2x-1}. \qquad (7)$$

Then the expectation and variance of $\hat{\alpha}_i$ are (plugging in Eqs. (5) and (6))

$$\mathbf{E}(\hat{\alpha}_i) = \frac{\mathbf{E}(u_i) - (1-x)}{2x-1} = \frac{i}{E} = \alpha_i, \ \mathbf{Var}(\hat{\alpha}_i) = \frac{\mathbf{Var}(u_i)}{(2x-1)^2} = \frac{x(1-x)}{E(2x-1)^2}. \qquad (8)$$

Therefore, $\hat{\alpha}_i$ is an unbiased estimate of $\alpha_i$ so that the mean-squared error is just the variance and the root-mean-squared error is the standard deviation, as stated by Eq. (2).

### Pseudo code for the simulations in Fig. 2
To better explain the simulations involved in Fig. 2, we provide the pseudo code to implement the simulations. The essential idea is that in the simulations, we only need to specify the pairwise accuracy $x$ to obtain the restored sequence for a fine-grained ground-truth sequence of edges, i.e., no need to actually pass through an ensemble model to obtain the generation order of each edge pair.

**Algorithm 1.** Pseudo code to compute $D_i/E$ to plot Fig. 2b, c
   **Inputs:** number of edges $E$, pairwise accuracy $x$, number of repetitions $R$.
   Assuming that the ground-truth sequence of edges $\{e_1, \ldots, e_E\}$ is $\boldsymbol{\alpha} = (1, 2, \ldots, E)$, form the set of all edge pairs $\mathbb{S} = \{(e_i, e_j) : i, j = 1, \ldots, E, i < j\}$, then $|\mathbb{S}| = E(E-1)/2$. Let $M = \lfloor |\mathbb{S}| * x \rfloor$.
   **for** rep = 1 to $R$ **do**
       **Step 1:** Randomly select $M$ pairs from $S$ and assign the correct generation order to them; the remaining $|\mathbb{S}| - M$ pairs are assigned the wrong order.
       **Step 2:** Apply the ranking algorithm (i.e., Borda count) on the pairwise orders from Step 2 to get the restored sequence $\hat{\boldsymbol{\alpha}} = (\hat{\alpha}_1, \hat{\alpha}_2, \ldots, \hat{\alpha}_E)$.
       **Step 3:** Compute $D_i$ as $D_i = i - \hat{\alpha}_i$ for $i = 1, 2, \ldots, E$.
   **end for**

**Algorithm 2.** Pseudo code to compute $D_i/E$ corresponding to "Real Data" in Fig. 2d, e
   **Inputs:** a coarse-grained ground-truth sequence $\boldsymbol{\alpha}$, an ensemble model, number of repetitions $R$.
   Let $n$ be the number of snapshots in $\boldsymbol{\alpha}$ and $l_k$ be the number of edges in the $k$th snapshot, then $\boldsymbol{\alpha} = (1, \ldots, 1, 2, \ldots, 2, \ldots, n, \ldots, n)$ and $\sum_{k=1}^{n} l_k = E$, where $E$ is the length of $\boldsymbol{\alpha}$.
       **Step 1:** Obtain the restored sequence $\hat{\boldsymbol{\alpha}} = (\hat{\alpha}_1, \hat{\alpha}_2, \ldots, \hat{\alpha}_E)$ by passing through our ensemble mod- el and ranking algorithm. Then $\hat{\alpha}_1 \neq \hat{\alpha}_2 \neq \cdots \neq \hat{\alpha}_E$ and $\hat{\alpha}_i \in \{1, 2, \ldots, E\}$.
   **for** rep = 1 to $R$ **do**
       **Step 2:** Randomly assign fine-grained order to edges within the same snapshot to generate an intermediate sequence $\boldsymbol{\alpha}^* = (\alpha_1^*, \alpha_2^*, \ldots, \alpha_E^*)$, i.e., $\alpha_1^* \neq \alpha_2^* \neq \cdots \neq \alpha_E^*$, $\alpha_i^* \in \{1, \ldots, l_1\}$ for $i = 1, \ldots, l_1$, $\alpha_i^* \in \{l_1 + 1, \ldots, l_1 + l_2\}$ for $i = l_1 + 1, \ldots, l_1 + l_2, \cdots$, and $\alpha_i^* \in \{l_1 + \cdots + l_{n-1} + 1, \ldots, E\}$ for $i = l_1 + \cdots + l_{n-1} + 1, \ldots, E$.
       **Step 3:** Compute $D_i$ as $D_i = \alpha_i^* - \hat{\alpha}_i$ for $i = 1, 2, \ldots, E$.
   **end for**

**Algorithm 3.** Pseudo code to compute $D_i/E$ corresponding to "Simulation" in Fig. 2d, e
   **Inputs:** a coarse-grained ground-truth sequence $\boldsymbol{\alpha}$, pairwise accuracy $x$, number of repetitions $R$.
   **for** rep = 1 to $R$ **do**
       **Step 1:** Randomly assign fine-grained order to edges within the same snapshot to generate an intermediate sequence $\boldsymbol{\alpha}^*$ as Step 2 in Algorithm 2.
       **Step 2:** Obtain the restored sequence $\hat{\boldsymbol{\alpha}}$ as Step 1–2 in Algorithm 1.
       **Step 3:** Compute $D_i$ as $D_i = \alpha_i^* - \hat{\alpha}_i$ for $i = 1, 2, \ldots, E$.
   **end for**

### The linear transformation in transfer learning
The key to a successful transfer is to find a projection between Network $A$ and $B$ such that the low-dimensional vector representations of the

corresponding nodes in the two networks are as similar as possible. There are different ways to establish a corresponding relationship between the nodes of Network $A$ and $B$, here we consider the quantiles of the degrees of the nodes as an illustrating example. Thus, we first sort the nodes in both networks by their degrees in descending order and obtain the matrices consisting of vectors of the ordered nodes for both networks, denoted by $\mathbf{H_A}$ and $\mathbf{H_B}$. Then our goal is to find a linear transformation $\mathbf{L}$ such that $||\mathbf{H_B L} - \mathbf{H_A}||$ is minimized. By the least squares method, we obtain

$$\mathbf{L} = (\mathbf{H_B^\top H_B})^{-1}\mathbf{H_B^\top H_A}. \tag{9}$$

### Reporting summary

Further information on research design is available in the Nature Portfolio Reporting Summary linked to this article.

## Data availability

The network data used in this study are available at https://github.com/yijiaozhang/evolution_restore[55]. The source data generated in this study are provided in the Source Data file. Source data are provided with this paper.

## Code availability

The code for this study are available at https://github.com/yijiaozhang/evolution_restore[55].

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

## Acknowledgements

The authors would like to thank Bing-Yi Jing, Ziwei Dai, Meizhen Zheng, Hongxin Wei, and Guanhua Chen for many helpful discussions. This work is supported by the National Natural Science Foundation of China under Grants No. T2350710802, 12101294, 12275118 and 11931019. This research is supported in part by NUS AcRF Grant A-0004550-00-00 and the National Research Foundation, Prime Minister's Office, Singapore, and the Ministry of Communications and Information, under its Online Trust and Safety (OTS) Research Programme (MCI-OTS-001). Any opinions, findings and conclusions or recommendations expressed in this material are those of the author(s) and do not reflect the views of National Research Foundation, Prime Minister's Office, Singapore, or the Ministry of Communications and Information.

## Author contributions

Y.H. conceived the project. Y.-J.Z., C.X., J.S., J.X., L.F., T.Z.,and Y.H. designed the project. J.W. and J.L. performed experiments and numerical modeling. Y.-J.Z., C.X., L.F. and Y.H. wrote the manuscript.

## Competing interests

The authors declare no competing interests.
