## [Peer Review File · Nature Communications]

REVIEWER COMMENTS

Reviewer #1 (Remarks to the Author):

Please find the attached file.

Reviewer #2 (Remarks to the Author):

`\begin{document}`

This paper introduces a combination of machine learning techniques that, apparently, can reconstruct the temporal sequence of edge appearances in real networks. The problem is intriguing, and the results presented in the paper seem promising. However, as I will elucidate below, I have several doubts about the technical aspects of the manuscript, which hinder my recommendation for its publication in its current form.

`\begin{itemize}`

`\item`

I have some problems with the authors' choice of \mathcal{E} as a measure of performance. The authors state in the manuscript that

{\it Equation (2) shows that the overall error of the restored edge sequence is inversely proportional to the square root of the number of edges, suggesting that our approach has a huge advantage for networks with a rich number of edges. In other words, when the number of edges is large enough, we only need a machine learning model with accuracy slightly better than random guess for predicting the relative generation order of any two edges (i.e., $x > 0.5$) to make the overall error small. This is a really nice property and consistent with the results shown in Fig. 2b.}

I do not truly understand the rationale of this paragraph. The fact that \mathcal{E} is small for a large number of edges is a characteristic of the error measure, not of the inference method itself. In fact, a method with performance very close to random could result in very small values of \mathcal{E} even if the inference is essentially random. Why are the authors not using Kendall's tau instead? Kendall's tau

is a widely accepted measure for assessing the correlation between two ordered lists of objects, which would be ideal for comparing real and inferred sequences.

\item

Related to the point above, any measure of error will be strongly influenced in the case of type A networks (those with partial temporal information) just because the maximum deviation between the inference and the real appearance times are bounded by the size of snapshots. In fact, for several real networks (like in collaboration networks), the fraction of known ordered pairs of edges is close to 100% so that, in those cases, there is no much to be inferred. Why are the authors including them then?

\item

Also related to the two points above, I did not see any real-world example of reconstruction of type B networks, those for which only the final network is available. Why the authors did not report any example at this respect?

\item

The method is designed to work only for growing networks. However, there are many different ways a network can evolve. There are networks that are dynamic (like the Internet) that grow but also reorganize their edges over time, with links appearing and disappearing dynamically. There are also systems that grow by differentiation/specialization mechanisms, like explained in PNAS 118(21), e2018994118 (2021). What is the expected outcome of the inference method in these types of networks?

\item

The method, as it is now defined, needs to input the final number of nodes and edges. Can the method be used to project to the future and predict the emergence of new nodes and edges?

\item

A minor issue, the accuracy ϵ is not defined in the main manuscript.

\item

As a final remark, I would like to share a personal thought. As is often the case with machine learning methods, I find myself grappling with the feeling that I don't truly comprehend why the method works. For instance, what is the rationale behind embedding edges in the first step of the method, and what type of information do these embeddings capture? How do neural networks effectively utilize these embeddings to generate meaningful results? However, I believe that this essentially boils down to the black box problem that arises in all machine learning applications, so I have no objections in this regard.

\end{itemize}

\end{document}

Response to Reviewers

Reviewer 1:

Summary

In this work, the authors apply ensemble graph machine learning methods to estimate the sequence in which the edges of an observed network are likely to have appeared.

While the basic tools used in this work are well-established (the chosen node embeddings, ensemble classification and transfer learning) they have been combined in an imaginative way, and their results appear sound and compelling. The selection of particular techniques, namely comparative paradigm neural networks, and Borda's rule for rank aggregation, are not especially well known. From this alone, it's clear the study is deeply researched, and will be of interest to anyone familiar with the methodology of graph machine learning.

Recognising that the tools themselves aren't novel, the main value of the work lies in its thoroughly researched applications. First, using a fungi PPI network, the authors demonstrate how one might apply their method to unlabelled data, achieving plausible results. Second, the authors then verify that inferred edge sequences reproduce often observed traits of growing networks. These include preferential attachment, modularity (in the sense of functional assortativity where proteins of similar function are more densely connected), degree assortativity, and local clustering. The inferred network evolution exhibits these properties convincingly. The third application is to show that using their method, the task of link prediction can be improved.

This is a detailed, substantial work that appears technically sound. A strength of the work is the breadth of data examined, spanning several complex network subdomains. Methods and SI are detailed and referenced judiciously throughout the main text.

We wish to thank the reviewer for an accurate summary of our paper, which highlights the reviewer's expertise on the subject. We are delighted to hear that the reviewer has grasped the key contribution of the paper, appreciates that our study is deeply researched, detailed and substantial. In the following, we provide point-by-point responses to each of the thoughtful comments that the reviewer shares with us.

Technical comments

I have little to say about the technical implementation - the authors carefully implement well researched tools from the graph machine learning toolkit. My comments therefore pertain to their application and interpretation.

We are very grateful for the reviewer's positive comments on our technical implementation.

Major comments

1. The authors claim that their method provides biologists with a novel tool to explore protein-protein interaction data. While I agree their reconstruction appears plausible, I

don't see how it is of concrete use. Is the reconstruction verifiable in any way? If biologists lack ground truth (i.e. paleontological) data, how much confidence can domain experts have in these reconstructions? I appreciate that this is a generic drawback of deep-learning techniques, which are difficult to interpret and assign confidence to. Nevertheless, this section is simply too brief for the application to be convincing, and I would need a clearer explanation of how the authors expect the construction to be used.

Thanks to the reviewer for raising these questions. We first explain our understanding on “lack of ground truth data”: we do not have the ground truth of the order in which each edge in the network is generated; only the ground truth of the coarse-grained snapshots of the edges are available. For networks where even coarse-grained snapshots are unavailable, transfer learning is a possible method for reconstruction, but the technique is not yet mature to be applied to real-world networks (see our response to the reviewer’s next question on transfer learning).

Then, we would like to address the reviewer’s concerns from the following three aspects:

1) Verification of the reconstruction through “cross-validation”.

For the case lacking ground-truth data, there is currently no recognized and effective verification method. **Due to the randomness of the restoration itself, (including the embedding and the training of the ensemble model), the results will be different every time we go through the reconstruction process.** This inspires us to verify the credibility of the reconstruction by “cross-validation” on these random results.

The idea of “cross-validation” is based on the following intuitive assumption: if the reconstruction process is reliable, then the random results should all be close to each other since they are all close to the ground truth; however, if the reconstruction process is unreliable, then the random results should be far away from each other since they are far away from the ground truth in different directions. Based on such an intuitive assumption, we could verify the credibility of the reconstruction by examining the difference between the random results. Specifically:

- I. Examine the difference between results from multiple trainings using the same machine learning model. We find that as more edge pairs are used to train the model, the outputs from the same model become closer (Fig. R1a-c).
- II. Examine the difference between results of different machine learning models. We find that as more edge pairs are used to train the model, the outputs of the different models also become closer (Fig. R1d-f).

Both points suggest that the reconstruction process is reliable. Note that when measuring the difference between two restored edge sequences, the error \mathcal{E} defined by Eq. (1) in the main text is used by treating one of the two restored edge sequences as the ground-truth sequence. This is the reason why the \mathcal{E} in Fig. R1 and Fig. R2 are larger than those from the theoretical results in Fig. 2b of the main text. As we are not comparing each random result with the underlying ground truth, but comparing two random results, the error here would be approximately twice the error of the theoretical results.

Figure R1. Verification of the reconstruction through “cross-validation” based on the PPI network for Fungi, Bacteria, and Human. **a-c** Average difference between restored edge sequences from multiple trainings of the same machine learning model. For a fixed training ratio, each machine learning model is trained 10 times to obtain 10 restored edge sequences. Then calculate the \mathcal{E} between each pair of restored edge sequences and get $10 \times (10 - 1)/2 = 45$ errors. Each point in a-c is the average of the 45 errors. **d-f** Average difference between restored edge sequences from different machine learning models (i.e., using different edge representation methods). When comparing two models, for a fixed training ratio, each model is trained 10 times so that 10 restored edge sequences are obtained for each model. Then calculate the \mathcal{E} between each pair of restored edge sequences from the two models and get $10 \times 10 = 100$ errors. Each point in d-f is the average of the 100 errors. The gray dashed line represents the average difference between two random edge sequences. Please refer to Fig. S1 in the SI for an illustration of the models, i.e., the Node2Vec, DeepWalk, and SDNE correspond to the first three base models to be ensembled in Fig. S1.

However, the results based on the PPI network for Fruit Fly and Worm, as shown in Fig. R2 below, do not follow the same pattern, especially when comparing different models (see Fig. R2c-d). Possible reasons are that the number of edges of the PPI network for Fruit Fly and Worm are relatively small compared to other PPI networks, and the pairwise accuracy of the models trained on these two PPI networks are not as high as those on other PPI networks (see Tab. R1). Therefore, we acknowledge that the restoration results for Fruit Fly and Worm may be suboptimal. However, this does not diminish the significance of our paper's contribution since the restoring results are reliable for other PPI networks. We appreciate the question raised by the reviewer, as it has provided us with a deeper understanding of the restoration results, and we have added some discussions to the updated SI.

Figure R2. “Cross-validation” based on the PPI network for Fruit Fly, and Worm. a-b Same with a-c in Figure R1. **c-d** Same with d-f in Figure R1.

Table R1. Basic information of the PPI networks and the accuracy of different methods. From left to right, we report network name, number of nodes N , number of edges E at the final snapshot, the pairwise accuracy x of CPNN base models with DeepWalk, Node2vec, LINE, Struct2vec, SDNE, and the ensemble model. The average accuracy over 100 simulations and the standard deviation (in the parentheses) are both reported. For each network, the highest accuracy is highlighted in bold face.

PPI Network	N	E	DeepWalk	Node2Vec	LINE	Struct2Vec	SDNE	Ensemble
Fungi	2144	6000	0.856 (0.005)	0.851 (0.008)	0.691 (0.012)	0.756 (0.009)	0.819 (0.007)	0.877 (0.007)
Bacteria	873	2321	0.878 (0.022)	0.878 (0.024)	0.781 (0.012)	0.846 (0.011)	0.878 (0.007)	0.909 (0.010)
Human	1891	2840	0.878 (0.008)	0.885 (0.007)	0.776 (0.010)	0.738 (0.015)	0.819 (0.011)	0.897 (0.008)
Fruit Fly	461	598	0.790 (0.021)	0.801 (0.018)	0.771 (0.028)	0.712 (0.017)	0.714 (0.022)	0.797 (0.029)
Worm	485	438	0.697 (0.022)	0.700 (0.029)	0.693 (0.015)	0.694 (0.028)	0.707 (0.018)	0.742 (0.025)

2) Verification on networks with ground-truth.

To further validate the reliability of our reconstruction results, verifications on synthetic networks with complete ground-truth are conducted. We artificially mask the fine-grained edge sequence of a network to be coarse-grained and train our model on the coarse-grained edge sequence. The resulting pairwise accuracy x remains high as shown in the SI Section 5.4 and summarized in Tab. R2. Specifically, we mask the edge sequence of a BA network model with 1991 snapshots, into 3, 5, and 7 snapshots, resulting in pairwise accuracy $x = 0.8257, 0.8280, \text{ and } 0.8496$, respectively. These values are considerably high compared to $x = 0.8938$ obtained using all 1991 snapshots. Therefore, we believe that our method is still effective when there is only coarse-grained ground-truth available.

Table R2: Accuracy of the ensemble model trained on BA networks with different time granularity.

Number of snapshots S	3	5	7	1991
Pairwise accuracy x	0.8257	0.8280	0.8496	0.8938

3) Explanation on the expected applications of the reconstruction.

In addition to the applications mentioned in the main text, we expect to have potentially important applications in the following fields.

- a. One of the most promising fields is **biomedicine**. As shown in Fig. R3 below, in biomedical research, protein interaction networks, gene similarity networks, drug component association networks, and disease networks can be coupled to predict whether an old treatment that have passed Phase III clinical trials would be effective on other diseases, or to predict new compounds with potential therapeutic effects on a certain disease [1]. These are essentially link prediction problems. As we are able to restore the edge formation history of the protein interaction networks and other related networks in this prediction system, it is expected to greatly improve the link prediction accuracy.
- b. Another potential field is **ecology**. Ecological networks are complex networks consist of interactions between species with evolution being the most important feature. By restoring the historical evolution trajectory of ecological networks, we can reveal the evolutionary rules and trends, predict the pattern of interactions between species, and assess the stability of ecosystems. This helps ecologists better understand and manage the ecosystems, forecast the risk of species extinction and invasion, and evaluate the resilience of ecosystems.
- c. The third application we think of is a general application in studying complex networks. Complex systems' evolution mechanisms are critical in understanding and predicting the systems' behavior. However, due to the lack of detailed historical observation data, we only know some simple evolution mechanisms and the corresponding network structure characteristics expressed by the mechanisms, such as the preferential attachment rule and the power-law degree distribution. Our reconstruction enables data-driven modeling on the evolution mechanisms of networked complex systems. This helps reveal complex systems' self-organizing characteristics and phase transition behavior, and further provides better methods to model complex systems. Fig. 3 and Fig. 4 in the main text are our initial explorations in this direction.
- d. One step further, if we can infer not only the edge generation order, but also the time interval between the edge generation of complex protein networks, then we are able to see the speed of network evolution. This may enable us to observe the peak (the highest evolution speed) and time point of species explosion. This is just a very preliminary idea, but it is really interesting to think about it. We will try to implement the idea in subsequent follow-up research.

Figure R3. An illustration of Disease-Protein-Drug network.

2. The authors suggest that network data that is insufficiently timestamped might be approached using transfer learning from the 'same domain.' Since this is an applied work that the authors believe to be directly useful, I would expect more guidance as to the choice of related domain data. For example could a well labelled fungi PPIN be used to reconstruct an unlabelled bacteria network? How about a fruit fly, or a human?

We thank the reviewer for the comment. Frankly speaking, our current transfer learning technique can only be implemented on artificially synthesized networks with similar generation mechanisms. For real-world networks with limited historical information, we have tried several widely used transfer learning techniques (including training a model on Network A and then finetuning it on a small labeled dataset of Network B to reconstruct Network B), but the results are not satisfactory. However, since the potential applications of transfer learning is of significant value, we retain the results of transfer learning on artificial synthetic networks in the main text.

Our group has been working hard to solve the problem of transfer learning on real-world networks and has conducted extensive discussions with experts in the field. Our current understanding about the difficulty of the problem is summarized as follows:

- 1) For insufficiently timestamped real-world networks, the key for transfer learning to be successful is that the way of embedding nodes or edges of different networks should be the same. That is, nodes or edges with similar topological characteristics in different networks should be close after being mapped to the embedding space. However, there is inherent randomness in the embedding algorithms currently applied, e.g., DeepWalk and Node2Vec. Consequently, when applying an embedding algorithm to different networks, the embedding spaces are not properly aligned. To achieve successful transfer learning, this is the first problem that must be solved.
- 2) For insufficiently timestamped real-world networks, the essential idea of transfer learning is to borrow information from other networks. Inspired by the pre-trained language models which significantly improved the performance of various natural language processing tasks, we need to design a graph neural network model pre-trained on real-world networks with rich timestamps. However, research on pre-trained graph neural network models in its early stages, and there is no generally accepted method currently. This is the second problem to be solved to implement transfer learning.

We will continue to explore transfer learning in real-world networks without sufficient timestamps in subsequent research.

3. A more serious discussion of the limitations of this method would be appreciated. This is currently confined to two sentences in the conclusion, and I don't believe a proper effort has been made here. One of the stated limitations (that the method only provides the order) I don't consider a substantial limitation at all.

We appreciate the reviewer's comment which remind us to reexamine the limitations of our method. We have added a discussion of limitations in the conclusion, especially addressing the difficulties in transfer learning.

Minor comments

1. It wasn't clear to me when transfer learning was required in the applications, and when there was sufficient labelling to train a network directly. For example, did the PPI network in 2.4.1 contain edge timestamps?

We thank the reviewer for raising this question. The PPI network in Section 2.4.1 contain the timestamps for part of the edges so that supervised learning has been applied to obtain the restored edge sequence.

To answer the reviewer's question "when there was sufficient labelling to train a network directly", we show how the pairwise accuracy of our ensemble model on test data changes when the percentage of edge pairs used for training are really small (see Fig. R4 below). The training ratio refers to the ratio of the number of edge pairs in the training set to the total number of edge pairs in the network. For most real-world networks in our study, a pairwise accuracy around 0.7 can be obtained when the training ratio is 0.001. For example, for a network with 1,000 edges, there are $1000 \times 999/2 = 499500$ edge pairs in total; if we know the relative order of $499500 \times 0.001 \approx 500$ edge pairs, then we can try to train an ensemble model directly.

Figure R4. The test accuracy of the ensemble model as a function of the percentage of edge pairs used for training. This is just an enlarged version of Fig. 2a in the main text which focus on the training ratio between 0.00 and 0.045.

2. Since this is an applied paper, it would be nice to share the code, data or both, or at least credit the libraries and software you've used.

The reviewer is absolutely right that the code and data should be shared. All code, data, and readme files will be made publicly available once the paper is accepted for publication [3]. We have also added the Data and Code Availability section in the main text.

3. Since the authors study local clustering in the sense of Watts and Strogatz, a mention of the average path lengths would be of interest. Recovering the small-world effect (high local clustering, but short average path length) using this method would be striking, on top of the already impressive topological results.

We appreciate the reviewer’s comment which inspire use to study the small-world effect in the restored network evolution processes. We explore how the average path length changes as the network evolves (i.e., the number of edges increases), the results are shown in Fig. R5g-i. It is shown that the results based on our restored edge sequences (blue solid lines) are closer to those based on the ground-truth data, and obtain shorter average path length and higher local clustering coefficients compared with those based on random sequences (red dashed line). These are consistent with the small-world effect in real-world networks.

Figure R5. Assortativity coefficient, local clustering coefficient, and average shortest path length for the restored evolution processes. The PPI network (Bacteria) (a, d, g), the World Trade Web (WTW) (b, e, h), and the Animal network (Weaver) (c, f, i) are selected. The yellow triangles represent results computed at the real snapshots of the networks. The blue solid lines and red dashed lines are results based on edge generation order by our restoring method and by random assignment, respectively. The pink dashed lines are results for networks generated assuming pure PA rule. Note that due to the presence of disconnected components during the evolution process of a network, the computation of average shortest path length only involve pairs of nodes that can be connected.

The results for other real-world networks are reported in the SI (Fig. S16-S18). Note that the subfigure **d** in Fig. 5 of the original main text was for PPI (Human), to keep consistency between different measures, it has been updated to be based on PPI (Bacteria). The results for PPI (Human) are moved to the SI instead.

4. In the same vain, in the manuscript I would specify that you mean 'local' clustering, since the triangle meaning might be confused with mesoscopic clustering, like modularity, which you also examine.

Thanks to the reviewer for pointing this out. We have changed “clustering” to “local clustering” in the manuscript to avoid confusion. Explanation and citation on “local clustering” have also been added when it first appears in the manuscript.

5. It is not clear to me how the 'best feature' is identified. In the SI the authors say they identify the node feature (e.g. edge betweenness, edge degree) that best predicts edge appearance order. How is this done exactly?

We are sorry that we didn't explain clearly enough in the last version of the manuscript. First, 11 features (e.g., edge betweenness, edge degree, etc., see SI Section 1.1 for a full list) are computed for each edge. Then for each edge pair, we make judgements about the edge generation order by comparing the value of each feature of the two edges. Finally, the “best feature” is identified as the one which has the highest accuracy in determining the relative generation order of edge pairs. In addition, we also construct a “collection feature” for each edge by combining the 11 features to an 11-dimensional vector and treat it as an embedding of the edge to be feed into a CPNN (comparative paradigm-based neural network) model, which is one of the base models in the ensemble model (see SI Fig. S1).

Editorial comments

The manuscript is extremely well written making it easy to read. Also, the figures and beautiful, and easy to interpret despite being dense in information. Nevertheless, I have some comments on how it might be improved.

We wish to thank the reviewer for his/her recognition of the writing and figure style of our paper. By adopting the reviewer's suggestions, we believe that the paper has significant improvement compared to the first submission.

Major comments

1. I found the some of the language used to describe the authors' contributions to be disturbing. For instance, they claim to perform precise network reconstruction 'for the first time,' that they have 'remarkably improved' link prediction, that the work is 'fundamental' and 'momentous.' This language only hurts the manuscript, and even raises suspicion among readers. For example, the authors state that the subdiscipline of link prediction has 'hit a roadblock' that their method overcomes. Such a claim is so strong that I doubt my capacity to evaluate it, only substantial further review by experts active in this problem can address such claims.

Thanks to the reviewer for this pertinent suggestion. After searching on Google Scholar, we found that no other research group has studied the problem of restoring the historical evolution process of networks. This was the reason why we used expressions such as “for the first time” and “hit a roadblock” in the last version of the manuscript. However, we agree with the reviewer and acknowledge that our knowledge may be limited and that our search may have missed some relevant studies, so “for the first time” has been removed and other expressions have been modified accordingly.

2. In many places, the language is needlessly vague. The authors refer to 'networks with a rich number of edges.' What does rich mean here? Does it simply mean a large number of edges? This is just one example, reading the manuscript I was overwhelmed by the use of adjectives.

We thank the reviewer for pointing this out. We are sorry that some descriptions in the manuscript were not clear enough. In the updated text, we have changed “networks with a rich number of edges” to “networks with a large number of edges” and carefully checked and revised other adjectives with ambiguous meanings.

3. I found some sections to suffer due to the terseness of the descriptions. In particular, in sections 2.1 and 2.2 I found myself constantly flipping between the manuscript, the methods and the SI (the latter is detailed, which is helpful, but very long as a result). It's still not entirely clear to me what the simulations in Fig. 2b, c, d and e involve. I would find an expanded section 2 to be helpful. Put simply, so much work was done by the authors, the detail in the main text doesn't do enough to describe it.

We are sorry that due to the word limit of the journal, we have to put many details into the Methods section and the SI, so that the main text only includes the basic ideas and main results of the proposed method. According to the instructions of *Nature Communications*, “the main text (not including Abstract, Methods, References and Figure legends) should be limited to 5,000 words” [4]. For Fig. 2b-e, the pseudo code to perform the simulations have been added to the Methods section, hopefully these, together with Fig. 2f and the figure caption, would make the simulation details clearer to the reviewer.

Minor comments

1. Subplot labels in Figures 1 and 2 could be made bold to be consistent with later figures

Thanks to the reviewer for this careful observation. We have made the revision in the updated manuscript.

2. Figure S12 in the SI is labelled e, I spent quite some time trying to find the corresponding figure to which it was related, as I assumed it was a snapshot of a bigger figure

We appreciate the reviewer’s effort for carefully going over the SI. Sorry that we made a mistake here and the “e” has been removed in the updated SI.

3. The caption of Figure 6 refers to the 'real generation order.' What is meant by 'real'? Do you mean ground truth?

The reviewer is right that “real” means “ground-truth” here, i.e., we have the ground-truth timestamps of some of the edges in the data. However, the ground-truth timestamps are coarse-grained. For example, there are only three snapshots in the raw data of the PPI (Fungi) so that the “real (or ground-truth) generation order” is $\alpha = (1, 1, \dots, 1, 2, 2, \dots, 2, 3, 3, \dots, 3)$. The relative generation order of edges from different snapshots can be identified, but that of edges within the same snapshots cannot. After restoration with our method, a fine-grained edge generation order can be obtained, e.g., $\hat{\alpha} = (2, 1, 3, 5, 7, \dots)$. And the results in Fig. 6 show that link prediction with the restored fine-grained order $\hat{\alpha}$ perform better than that with the ground-truth coarse-grained order α .

4. A number of typos in the bibliography will need correction, such as inconsistent periods after initials, lower cases in names and journals, as well as punctuation

We thank the reviewer for pointing this out. We have double-checked all references and corrected the typos.

Conclusion

This work is a strong contribution, and I believe it meets the high standards of Nature Communications. However, I would need to see it again after the authors attempt to address the concerns I describe above.

Once again, we would like to thank the reviewer for the positive comments and constructive suggestions on our work. Hope that our revisions would address the reviewer’s concerns adequately.

Reviewer 2:

This paper introduces a combination of machine learning techniques that, apparently, can reconstruct the temporal sequence of edge appearances in real networks. The problem is intriguing, and the results presented in the paper seem promising. However, as I will elucidate below, I have several doubts about the technical aspects of the manuscript, which hinder my recommendation for its publication in its current form.

We wish to thank the reviewer for a concise and precise summary of our paper. We are delighted to hear that the reviewer think that the problem is intriguing, and the results are promising.

1. I have some problems with the authors' choice of \mathcal{E} as a measure of performance. The authors state in the manuscript that

“Equation (2) shows that the overall error of the restored edge sequence is inversely proportional to the square root of the number of edges, suggesting that our approach has a huge advantage for networks with a rich number of edges. In other words, when the number of edges is large enough, we only need a machine learning model with accuracy slightly better than random guess for predicting the relative generation order of any two edges (i.e., $x > 0.5$) to make the overall error small. This is a really nice property and consistent with the results shown in Fig. 2b.”

I do not truly understand the rationale of this paragraph. The fact that \mathcal{E} is small for a large number of edges is a characteristic of the error measure, not of the inference method itself. In fact, a method with performance very close to random could result in very small values of \mathcal{E} even if the inference is essentially random. Why are the authors not using Kendall's tau instead? Kendall's tau is a widely accepted measure for assessing the correlation between two ordered lists of objects, which would be ideal for comparing real and inferred sequences.

Thanks to the reviewer for this profound comment. In the following, we would like to address the reviewer's concern on the overall error \mathcal{E} from two aspects.

1) By definition, \mathcal{E} does not naturally decrease as E increases since it has been normalized to the number of edges E . We demonstrate this through a simulation study: we randomly permute an edge sequence of a given length E and then calculate the overall error \mathcal{E} between the permuted sequence and the original sequence. The results at different values of E are shown in Fig. R6, indicating that \mathcal{E} remains at the same level as E increases. To be more specific, it is the standard deviation of \mathcal{E} which decreases as E increases, not the mean of \mathcal{E} . Through further theoretical derivation, we obtain that the theoretical error between a randomly permuted sequence and the original sequence is $1/\sqrt{6} \approx 0.408$, which is consistent with the simulation results in Fig. R6. Detailed theoretical derivation is provided below:

Let $\alpha = (\alpha_1, \alpha_2, \dots, \alpha_E) = (1, 2, \dots, E)$ be the original sequence, and $\hat{\alpha} = (\hat{\alpha}_1, \hat{\alpha}_2, \dots, \hat{\alpha}_E)$ be the randomly permuted sequence. The permuted order of edge i , i.e., $\hat{\alpha}_i$, follows the uniform distribution on $\{1, 2, \dots, E\}$. Then we consider the order normalized by E , when E is large, the normalized $\hat{\alpha}_i/E$ approximately follows the uniform distribution on $[0, 1]$ with probability density function $f(z) = 1, z \in [0, 1]$. For simplicity, we denote the normalized

order by α_i and $\hat{\alpha}_i$. So, the expectation of the normalized squared error of edge i , i.e., $D_i^2 = (\alpha_i - \hat{\alpha}_i)^2$ is

$$\mathbf{E}_{\hat{\alpha}_i}[D_i^2] = \int_0^1 (\alpha_i - z)^2 f(z) dz = \int_0^1 (\alpha_i - z)^2 dz = \alpha_i^2 - \alpha_i + \frac{1}{3} := h(\alpha_i).$$

Then we need to further average $\mathbf{E}_{\hat{\alpha}_i}[D_i^2] = h(\alpha_i)$ over all edges:

$$\frac{1}{E} \sum_{i=1}^E h(\alpha_i) \approx \int_0^1 h(\alpha_i) d\alpha_i = \int_0^1 \left(\alpha_i^2 - \alpha_i + \frac{1}{3} \right) d\alpha_i = \frac{1}{3} - \frac{1}{2} + \frac{1}{3} = \frac{1}{6}.$$

Therefore, the theoretical error is

$$\mathcal{E} = \lim_{E \rightarrow \infty} \sqrt{\frac{1}{E} \sum_{i=1}^E \mathbf{E}_{\hat{\alpha}_i}[D_i^2]} = \frac{1}{\sqrt{6}}$$

Figure R6. The overall error \mathcal{E} between an original edge sequence and a randomly permuted edge sequence as the number of edges E increases. Each blue circle represents the average overall error of 100 permutations and each error bar represents the standard deviation. The yellow horizontal line is the theoretical error.

- 2) Then we also compare our overall error \mathcal{E} with other widely accepted measures for assessing the correlation between two ordered sequences, including the Kendall's τ mentioned by the reviewer [5], as well as the Spearman's ρ [6]. We find that theoretically, \mathcal{E} has a monotonic functional relationship with the other measures, suggesting that they are essentially equivalent but differ in value.

a. The theoretical equivalence between \mathcal{E} and Kendall's τ .

By definition [5], Kendall's τ between two ordered sequences, e.g., α and $\hat{\alpha}$, is

$$\tau = 1 - \frac{2(\text{number of discordant pairs})}{\text{number of pairs}}.$$

The number of pairs is $\binom{E}{2} = E(E-1)/2$, denote the number of discordant pairs by K_d , then

$$K_d = |\{(i, j): i < j, (\alpha_i < \alpha_j \wedge \hat{\alpha}_i > \hat{\alpha}_j) \vee (\alpha_i > \alpha_j \wedge \hat{\alpha}_i < \hat{\alpha}_j)\}|,$$

We have shown in the manuscript that the expectation and variance of $\hat{\alpha}_i$ are

$$\mathbf{E}(\hat{\alpha}_i) = \alpha_i, \mathbf{Var}(\hat{\alpha}_i) = \frac{(1-x)x}{E(2x-1)^2}.$$

So, $\hat{\alpha}_i$ fluctuates around α_i and the range of fluctuation is proportional to

$$\sigma = \sqrt{\mathbf{Var}(\hat{\alpha}_i)} = \frac{\sqrt{(1-x)x}}{2x-1} \sqrt{\frac{1}{E}}.$$

Similarly, $\hat{\alpha}_j$ fluctuates around α_j and the range of fluctuation is proportional to σ . Therefore, the proportion of discordant pairs where $(\hat{\alpha}_i, \hat{\alpha}_j)$ demonstrates a different order with (α_i, α_j) , is proportional to σ :

$$\frac{K_d}{E(E-1)/2} \propto \sigma = \frac{\sqrt{(1-x)x}}{2x-1} \sqrt{\frac{1}{E}} \propto \mathcal{E}^{\text{theory}},$$

Therefore, we have that

$$\tau = 1 - \frac{2K_d}{E(E-1)/2} = 1 - b\mathcal{E}^{\text{theory}}.$$

This theoretical relationship is displayed in Fig. R7a (the yellow solid line), which is consistent with the simulation results (the blue dots). According to the simulation results, $b \approx 2.18$.

b. The theoretical equivalence between \mathcal{E} and Spearman's ρ .

By the definition of Spearman's ρ [6], we have

$$\rho = 1 - \frac{6 \sum_{i=1}^E D_i^2}{E(E^2 - 1)} = 1 - \frac{6E^2}{E^2 - 1} \times \frac{1}{E} \sum_{i=1}^E \left(\frac{D_i}{E}\right)^2 = 1 - \frac{6E^2}{E^2 - 1} \mathcal{E}^2.$$

When E is large, we have $E^2 \approx E^2 - 1$ so that

$$\rho = 1 - 6\mathcal{E}^2.$$

This theoretical relationship is displayed in Fig. R7b (the yellow solid line), which is also consistent with the simulation results (the blue dots)

Figure R7. The relationship between the overall error \mathcal{E} and Kendall's τ (a), Spearman's ρ (b). The yellow solid lines represent the theoretical relationships and the blue dots are the simulation results. The simulation is performed under the number of edges $E = 10000$ and the pairwise accuracy $x \in [0.51, 0.95]$. Each dot (the combination of E and x) is repeated 10 times.

In summary, \mathcal{E} would not decrease as the number of edges E increases and it is theoretically equivalent with other widely used measures like Kendall's τ and Spearman's ρ . The reason

why we choose \mathcal{E} as the measure of performance is that its physical meaning is more intuitive than other measures.

Interestingly, there is a similar formula in the field of localization accuracy in single-molecule microscopy [7]. The localization accuracy for a single molecule is also inversely proportional to the square root of the number of observed molecules. This is a really nice property and consistent with the results shown in Fig. 2b in the main text.

2. Related to the point above, any measure of error will be strongly influenced in the case of type A networks (those with partial temporal information) just because the maximum deviation between the inference and the real appearance times are bounded by the size of snapshots. In fact, for several real networks (like in collaboration networks), the fraction of known ordered pairs of edges is close to 100% so that, in those cases, there is no much to be inferred. Why are the authors including them then?

In many real-world networks, the fraction of edge pairs with known order is relatively small. For these networks with limited ground-truth data, only very little training data can be used to train the ensemble model. To better illustrate the effectiveness of our method on different network sizes, we also include several networks with a large number of edge pairs with known order to verify the effectiveness of our restoration method. We completely agree with the reviewer that restoring networks with almost 100% known edge generation order does not make much sense in practice, but they are useful for testing when testing the effectiveness and reliability of our method.

3. Also related to the two points above, I did not see any real-world example of reconstruction of type B networks, those for which only the final network is available. Why the authors did not report any example at this respect?

We thank the reviewer for the comment. Frankly speaking, our current transfer learning technique can only be implemented on artificially synthesized networks with similar generation mechanisms. For real-world networks with limited historical information, we have tried several widely used transfer learning techniques (including training a model on Network A and then finetuning it on a small labeled dataset of Network B to reconstruct Network B), but the results are unsatisfactory. However, since the potential applications of transfer learning is of great value, we still keep the results of transfer learning on artificial synthetic networks in the main text.

Our group has been working hard to solve the problem of transfer learning on real-world networks and has conducted extensive discussions with experts in the field. Our current understanding about the difficulty of the problem is summarized as follows:

- 1) For insufficiently timestamped real-world networks, the key for transfer learning to be successful is that the way of embedding nodes or edges of different networks should be the same. That is, nodes or edges with similar topological characteristics in different networks should be close after being mapped to the embedding space. However, there is inherent randomness in the embedding algorithms currently applied, e.g., DeepWalk and Node2Vec. Consequently, when applying an embedding algorithm to different networks, the embedding spaces are not properly aligned. To achieve successful transfer learning, this is the first problem that must be solved.

- 2) For insufficiently timestamped real-world networks, the essential idea of transfer learning is to borrow information from other networks. Inspired by the pre-trained language models which significantly improved the performance of various natural language processing tasks, we need to design a graph neural network model pre-trained on real-world networks with rich timestamps. However, research on pre-trained graph neural network model has just started, and there is no generally accepted method currently. This is the second problem to be solved to implement transfer learning.

We will continue to explore transfer learning in real-world networks in subsequent research.

4. The method is designed to work only for growing networks. However, there are many different ways a network can evolve. There are networks that are dynamic (like the Internet) that grow but also reorganize their edges over time, with links appearing and disappearing dynamically. There are also systems that grow by differentiation/specialization mechanisms, like explained in PNAS 118(21), e2018994118 (2021). What is the expected outcome of the inference method in these types of networks?

As discussed in the conclusion of the main text, there are some limitations in our work. A limitation of our method is that it does not take into account the situation where edges disappear as the network evolves. For the geometric branching growth (GBG) model in the PNAS paper mentioned by the reviewer [8], it allows node splitting when generating a network which is similar to the PPI networks used in our study. We contacted Professor Zheng, the first author of the PNAS paper, and he kindly shared the network generated by the GBG model with us. Therefore, we can try our method on the network and present the results. First, a data preprocessing step (see SI Sec. 3.1, the preprocessing method is the same as PPI networks) is applied to identify the edge generation timestamps of the network. Then, our ensemble model is trained and applied to restore the edge generation order. The results are shown in Fig. R8, indicating that our ensemble model works for a network generated by the GBG model and the performance is comparable to those on real-world networks.

Figure R8. Performance of different models on the network generated by the GBG model. Parameter values of the network generated by the GBG model: branch rate = 0.2, number of snapshots = 5, number of nodes = 1026, number of edges = 5344.

5. The method, as it is now defined, needs to input the final number of nodes and edges. Can the method be used to project to the future and predict the emergence of new nodes and edges?

Thanks to the reviewer for raising this question. Our method is able to predict the emergence of new edges between existing nodes. The link prediction results displayed in Fig. 6 in the main text are exactly predicting future edges. However, we cannot forecast the emergence of new nodes.

6. A minor issue, the accuracy x is not defined in the main manuscript.

Thanks to the reviewer for carefully checking our paper. In the main text, we actually defined x by stating “the test accuracy x of the ensemble model in predicting the relative generation order of any two edges...”. We are sorry that it is not clear enough to the readers. To be more specific, x is the number of edge pairs with correctly predicted generation order divided by the total number of edge pairs in the test set. We have emphasized the meaning of x in the updated manuscript when it first appears.

7. As a final remark, I would like to share a personal thought. As is often the case with machine learning methods, I find myself grappling with the feeling that I don’t truly comprehend why the method works. For instance, what is the rationale behind embedding edges in the first step of the method, and what type of information do these embeddings capture? How do neural networks effectively utilize these embeddings to generate meaningful results? However, I believe that this essentially boils down to the black box problem that arises in all machine learning applications, so I have no objections in this regard.

Thanks to the reviewer for these insightful and open questions, on which we would love to share our thoughts.

1) The rationale behind embedding and the information they capture.

In the first step of our method, the vector representations of the edges are obtained by learning the embedding of the nodes. This step is necessary because only by representing the edges as vectors can they be input into the machine learning model for training.

The essence of node/edge embedding is to make nodes/edges with higher similarity in topological structure closer in the embedding space. However, topological similarity is a quantity that is difficult to be defined clearly, and it may vary depending on the downstream tasks. Different embedding methods capture network structure with different emphases. For example, Node2Vec and DeepWalk try to bring nodes that appear close and frequently in the random walk sequence close in the embedding space, thereby capturing local information of the network structure.

It should be noted that the embedding methods used in our work does not rely on downstream tasks, i.e., the same set of edge embeddings can be used to train models for edge sequence reconstruction, link prediction, as well as community detection. There are many embedding methods that do rely on downstream tasks, such GNN [9], GCN [10], etc., which are not used in this work. We will explore the applications of these methods in subsequent research.

2) How do neural networks use the embeddings?

The prerequisite for neural networks to be able to restore the evolution history of the network is that the topological structure information of the edges contains information on time. Taking the BA network as an example, by the generation mechanism of the BA network, we know that nodes with larger degrees typically join the network earlier, so the edges linking two nodes with larger degrees usually appear earlier than those linking two nodes with smaller degrees. Therefore, the topological structure of a network contains information on edge generation time.

However, there are usually many short loops in the topological structure of a network, and the correlations between nodes are complex, making it difficult to theoretically identify whether there is a single structural feature that has a simple relationship with time. In fact, our results show that none of the 11 indices (e.g., edge betweenness, edge degree, etc., see SI Sec. 5.3 for a full list) can be used alone to effectively infer the edge generation sequence across different real-world networks (see Tab. R3 below, which is Tab. S3 in the SI, some indices are effective on some of the networks but not on others). But when the 11 indices are integrated to a vector and input to train a neural network model, the results are good for most real-world networks (see the last column of Tab. R3). This suggests that the neural network models are able to find a complex function that captures the higher-order relationships between the structural features and edge generation time, which we cannot obtain intuitively.

In summary, in our work, neural networks are applied to establish complex mappings from the vector representations of the edges to the edge generation sequences, which is consistent with the way machine learning technologies are applied to solve classical problems.

Table R3. Test accuracy of the CPNN model with single edge index. From left to right, we report network name, results of edge betweenness (BN), edge degree (DEG), common neighbor (CN), edge clustering coefficient (CC), edge strength (STR), resource allocation index (RA), Adamic-Adar index (AA), preferential attachment index (PA), local path index (LP), edge PageRank (PR), edge k-shell (KS), and collection feature (Collection). The value in the table represent the average accuracy and its standard deviation from 100 simulations. For each network, the best result is highlighted in bold face.

Network name	BN	DEG	CN	CC	STR	RA	AA	PA	LP	PR	KS	Collection
Fungi	0.62 ± 0.00	0.61 ± 0.01	0.62 ± 0.06	0.62 ± 0.01	0.61 ± 0.06	0.63 ± 0.04	0.64 ± 0.04	0.60 ± 0.03	0.60 ± 0.07	0.53 ± 0.01	0.62 ± 0.00	0.70 ± 0.01
Human	0.58 ± 0.04	0.56 ± 0.03	0.55 ± 0.00	0.56 ± 0.03	0.57 ± 0.02	0.56 ± 0.02	0.56 ± 0.02	0.56 ± 0.03	0.52 ± 0.02	0.52 ± 0.08	0.51 ± 0.02	0.70 ± 0.01
Fruit Fly	0.53 ± 0.05	0.60 ± 0.05	0.52 ± 0.02	0.50 ± 0.01	0.49 ± 0.01	0.50 ± 0.01	0.49 ± 0.02	0.57 ± 0.05	0.54 ± 0.03	0.58 ± 0.06	0.51 ± 0.01	0.64 ± 0.03
Worm	0.53 ± 0.03	0.55 ± 0.02	0.51 ± 0.01	0.51 ± 0.01	0.51 ± 0.01	0.51 ± 0.01	0.51 ± 0.01	0.54 ± 0.03	0.51 ± 0.03	0.54 ± 0.02	0.52 ± 0.01	0.61 ± 0.01
Bacteria	0.59 ± 0.03	0.59 ± 0.04	0.73 ± 0.08	0.69 ± 0.06	0.69 ± 0.10	0.74 ± 0.01	0.70 ± 0.10	0.70 ± 0.07	0.76 ± 0.01	0.49 ± 0.01	0.75 ± 0.08	0.82 ± 0.01
WTW	0.64 ± 0.07	0.76 ± 0.00	0.71 ± 0.14	0.54 ± 0.02	0.67 ± 0.11	0.74 ± 0.12	0.77 ± 0.09	0.79 ± 0.10	0.81 ± 0.00	0.65 ± 0.00	0.73 ± 0.12	0.87 ± 0.00
Complex Networks	0.50 ± 0.01	0.50 ± 0.01	0.52 ± 0.01	0.51 ± 0.01	0.52 ± 0.01	0.55 ± 0.02	0.54 ± 0.02	0.51 ± 0.01	0.51 ± 0.01	0.50 ± 0.00	0.51 ± 0.01	0.58 ± 0.01
Chaos	0.55 ± 0.00	0.52 ± 0.01	0.51 ± 0.01	0.53 ± 0.01	0.53 ± 0.02	0.50 ± 0.01	0.49 ± 0.01	0.52 ± 0.00	0.49 ± 0.01	0.53 ± 0.01	0.52 ± 0.01	0.57 ± 0.00
Fluctuations	0.54 ± 0.01	0.53 ± 0.01	0.51 ± 0.01	0.55 ± 0.00	0.53 ± 0.01	0.51 ± 0.01	0.51 ± 0.01	0.53 ± 0.01	0.50 ± 0.00	0.54 ± 0.00	0.51 ± 0.00	0.56 ± 0.01
Interfaces	0.54 ± 0.00	0.52 ± 0.01	0.54 ± 0.02	0.53 ± 0.00	0.55 ± 0.00	0.53 ± 0.01	0.55 ± 0.01	0.52 ± 0.01	0.55 ± 0.02	0.51 ± 0.01	0.54 ± 0.03	0.59 ± 0.00
Phase Transitions	0.56 ± 0.03	0.51 ± 0.01	0.54 ± 0.02	0.55 ± 0.02	0.57 ± 0.00	0.54 ± 0.02	0.55 ± 0.00	0.50 ± 0.00	0.54 ± 0.01	0.55 ± 0.01	0.54 ± 0.02	0.60 ± 0.00
Thermodynamics	0.56 ± 0.02	0.58 ± 0.05	0.51 ± 0.02	0.55 ± 0.04	0.54 ± 0.05	0.51 ± 0.03	0.50 ± 0.03	0.58 ± 0.01	0.50 ± 0.01	0.57 ± 0.04	0.50 ± 0.02	0.67 ± 0.02
Weaver	0.55 ± 0.02	0.56 ± 0.02	0.54 ± 0.01	0.50 ± 0.01	0.50 ± 0.01	0.50 ± 0.01	0.51 ± 0.01	0.56 ± 0.00	0.54 ± 0.02	0.53 ± 0.01	0.52 ± 0.02	0.67 ± 0.01
Ants	0.73 ± 0.08	0.72 ± 0.07	0.72 ± 0.07	0.59 ± 0.00	0.73 ± 0.08	0.74 ± 0.01	0.74 ± 0.01	0.62 ± 0.12	0.72 ± 0.07	0.56 ± 0.02	0.63 ± 0.04	0.76 ± 0.01
Airplane	0.68 ± 0.15	0.83 ± 0.11	0.53 ± 0.03	0.76 ± 0.09	0.69 ± 0.12	0.49 ± 0.03	0.51 ± 0.04	0.73 ± 0.15	0.51 ± 0.04	0.82 ± 0.11	0.63 ± 0.08	0.92 ± 0.02
Ferry	0.51 ± 0.02	0.58 ± 0.06	0.62 ± 0.01	0.62 ± 0.01	0.60 ± 0.05	0.60 ± 0.04	0.60 ± 0.03	0.56 ± 0.09	0.61 ± 0.07	0.54 ± 0.02	0.57 ± 0.04	0.68 ± 0.07
Coach	0.54 ± 0.04	0.74 ± 0.14	0.69 ± 0.07	0.68 ± 0.03	0.64 ± 0.08	0.65 ± 0.07	0.67 ± 0.06	0.78 ± 0.10	0.73 ± 0.12	0.74 ± 0.01	0.76 ± 0.01	0.77 ± 0.05

References

- [1] Huang, Kexin, et al. "Artificial intelligence foundation for therapeutic science." Nature chemical biology 18.10 (2022): 1033-1036.

- [2] Li, Michelle M., Kexin Huang, and Marinka Zitnik. "Graph representation learning in biomedicine and healthcare." *Nature Biomedical Engineering* 6.12 (2022): 1353-1369.
- [3] https://github.com/yijiaozhang/evolution_restore
- [4] <https://www.nature.com/ncomms/submit/article>
- [5] https://en.wikipedia.org/wiki/Kendall_rank_correlation_coefficient
- [6] https://en.wikipedia.org/wiki/Spearman%27s_rank_correlation_coefficient
- [7] Ober, Raimund J., Sripad Ram, and E. Sally Ward. "Localization accuracy in single-molecule microscopy." *Biophysical journal* 86.2 (2004): 1185-1200.
- [8] Zheng, Muhua, et al. "Scaling up real networks by geometric branching growth." *Proceedings of the National Academy of Sciences* 118.21 (2021): e2018994118.
- [9] Zhou, Jie, et al. "Graph neural networks: A review of methods and applications." *AI open* 1 (2020): 57-81.
- [10] Kipf, Thomas N., and Max Welling. "Semi-supervised classification with graph convolutional networks." *arXiv preprint arXiv:1609.02907* (2016).

REVIEWER COMMENTS

Reviewer #1 (Remarks to the Author):

The authors have carefully addressed my concerns and technical questions. I would be happy for publication to go ahead.

Some minor editorial comments below.

I would slightly revise the title as the current version sounds awkward. I asked chatGPT for a title that would sound more natural to a native English speaker and it produced "Reconstructing the evolutionary history of networked complex systems". This sounds a lot better to me. I also think "reconstructing" is more appropriate than "restoring" given the authors' methodology.

Some care still needs to be given to the reference section, e.g. britain instead of Britain in Ref. [35], world'networks instead of world' networks in Ref. [13], and Mark E. J. Newman instead of M. E. J. Newman to be consistent with the remainder of your reference section. I suspect these improvements can be automated by editors during publication.

Reviewer #2 (Remarks to the Author):

First of all, I apologize for taking so long to send my report. I recognize the effort the authors put in this new resubmission and I am prone to recommend the publication of the manuscript. However, there is still an issue that should be clarified before publication. I still have some concerns with Eq.~(2). In my previous report, I mentioned that the overall error used by the authors decreases with the number of edges. In their response, the authors argue that for reshuffled sequences of edges the overall error is constant. However, this corresponds to the random case $x=1/2$. In this case, Eq.~(2) has a divergence and, in fact, for a fixed number of edges, there is a value of x below which the error becomes larger than one. This already says that Eq.~(2) cannot be entirely correct. Thus, the first issue the authors should address is to determine the domain of validity of Eq.~(2).

My second concern, also related to Eq.~(2), is the fact that there is some "magic" going on here. Indeed, it is very strange that a method that could give partial results very close to random, let's say, $x=1/2+10^{-5}$, would lead to very small overall errors if E is large enough. So that using a very bad

method one ends up with a very accurate final result. I think this need some intuitive explanation in the manuscript, not just the bare formula in Eq.~(2). I think the problem lies (or my problem the first time I read the manuscript) in the fact that in the manuscript it is not very clear that the method has two parts, one where you get $\$x\$$ from pairwise results, and a second one where, using these pairwise results a ranking of edges is obtained. It is in this second step that the ``magic" happens and it should be made crystal clear in the manuscript. I recommend to put more emphasis here and give some more details of this second step.

Response to Reviewers

REVIEWER COMMENTS

Reviewer #1 (Remarks to the Author):

The authors have carefully addressed my concerns and technical questions. I would be happy for publication to go ahead.

Some minor editorial comments below.

I would slightly revise the title as the current version sounds awkward. I asked chatGPT for a title that would sound more natural to a native English speaker and it produced "Reconstructing the evolutionary history of networked complex systems". This sounds a lot better to me. I also think "reconstructing" is more appropriate than "restoring" given the authors' methodology.

Some care still needs to be given to the reference section, e.g. Britain instead of Britain in Ref. [35], world'networks instead of world' networks in Ref. [13], and Mark E. J. Newman instead of M. E. J. Newman to be consistent with the remainder of your reference section. I suspect these improvements can be automated by editors during publication.

Thanks to the reviewer for the suggestions on the paper title and reference section. We have made the revisions accordingly.

Reviewer #2 (Remarks to the Author):

First of all, I apologize for taking so long to send my report. I recognize the effort the authors put in this new resubmission and I am prone to recommend the publication of the manuscript. However, there is still an issue that should be clarified before publication. I still have some concerns with Eq.~(2). In my previous report, I mentioned that the overall error used by the authors decreases with the number of edges. In their response, the authors argue that for reshuffled sequences of edges the overall error is constant. However, this corresponds to the random case $x=1/2$. In this case, Eq.~(2) has a divergence and, in fact, for a fixed number of edges, there is a value of x below which the error becomes larger than one. This already says that Eq.~(2) cannot be entirely correct. Thus, the first issue the authors should address is to determine the domain of validity of Eq.~(2).

We really appreciate the reviewer for this deep-going comment which help us complete the work of this paper. In the previous version, we did not discuss the domain of validity of Eq. (2) in depth since the simulation results in Fig. 2b shows that our theory is correct. However, inspired by the reviewer, a further discussion on Eq. (2) is added.

Eq. (S24) in the Supplementary Information (SI) gives the variance of u_{ij} (the score of edge i when comparing the order of edge i and j in Borda count):

$$\text{Var}(u_{ij}) = \frac{x(1-x)}{E^2}.$$

Then, the variance and standard deviation of the total score of edge i , $u_i = \sum_{j=1, j \neq i}^E u_{ij}$, are

$$\text{Var}(u_i) = \sum_j \text{Var}(u_{ij}) = \frac{x(1-x)}{E},$$

$$\text{Std}(u_i) = \sqrt{\frac{x(1-x)}{E}}.$$

It has been shown in Section 6.1 of the SI that the scores of the edges are uniformly distributed over the interval $[1-x, x]$, which is of width $2x-1$. When $2x-1$ is too small, the variation range of u_i (i.e., $\text{Std}(u_i)$) would go beyond the interval, such that our derivation under this scenario is not valid. Therefore, our derivation is only valid under the condition that $\text{Std}(u_i)$ is much smaller than the width of the interval, i.e.,

$$\sqrt{\frac{x(1-x)}{E}} \ll 2x-1. \quad (\text{R1})$$

Let $x = 0.5 + \delta$, then the above inequality is easily satisfied when δ is not too small. So, we only consider the case when δ is very small so that $x \approx 0.5$. In this case $x(1-x) \approx 1/4$, and plugging this into Eq. (R1) we have

$$\delta \gg \frac{1}{4\sqrt{E}}. \quad (\text{R2})$$

Eq. (R2) is the domain of validity of Eq. (2), which quantifies how much is considered “slightly better than random guess” in the sentence “a machine learning model with accuracy slightly better than random guess” in the main text. Plugging Eq. (R2) into Eq. (2), we have

$$\mathcal{E}^{\text{theory}} = \frac{\sqrt{x(1-x)}}{2x-1} \frac{1}{\sqrt{E}} \ll 1.$$

Fig. R1 below illustrates the correctness of Eq. (2) when Eq. (R2) is satisfied: the simulation results are consistent with that from Eq. (2) when $\delta\sqrt{E} \gg 1/4$ (e.g., when $\delta\sqrt{E} > 2$, see fig.R1 b); when $\delta\sqrt{E} \ll 1/4$, the overall error of the simulation results is close to random guessing, i.e., $\mathcal{E} = 1/\sqrt{6}$. Considering that the number of edges in a network is typically large, the condition suggested by Eq. (R2) is not difficult to satisfy. For example, as Fig. R1a shows, when $E = 10,000$, a machine learning algorithm with pairwise accuracy $x = 0.5 + \delta (= 10^{-2}) = 0.51$ is quite enough for Eq. (2) to be valid.

Figure R1. The theoretical \mathcal{E} from Eq. (2) vs. the observed \mathcal{E} from simulations. **a** The relationship between δ and \mathcal{E} . **b** The relationship between $\delta\sqrt{E}$ and \mathcal{E} . In **a**, different colored solid lines are the theoretical results of Eq. (2) with different values of E and the dotted vertical lines show the corresponding positions of $\delta = 1/4\sqrt{E}$. In **b**, the black solid line is the theoretical results obtained from Eq. (2) (results with different E overlap) and the dotted vertical line shows the position of $\delta\sqrt{E} = 1/4$. Different colored symbols represent the simulation results with different values of E , where each point represents the average of 10 simulation runs. The error bars represent the standard deviation. The grey dashed line indicates the overall error of random guessing, i.e., $\mathcal{E} = 1/\sqrt{6}$.

My second concern, also related to Eq. (2), is the fact that there is some “magic” going on here. Indeed, it is very strange that a method that could give partial results very close to random, let's say, $x=1/2+10^{-5}$, would lead to very small overall errors if E is large enough. So that using a very bad method one ends up with a very accurate final result. I think this needs some intuitive explanation in the manuscript, not just the bare formula in Eq. (2). I think the problem lies (or my problem the first time I read the manuscript) in the fact that in the manuscript it is not very clear that the method has two parts, one where you get x from pairwise results, and a second one where, using these pairwise results a ranking of edges is obtained. It is in this second step that the “magic” happens and it should be made crystal clear in the manuscript. I recommend to put more emphasis here and give some more details of this second step.

The reviewer is absolutely right that our method has two steps, and it is in the second step that the “magic” happens. Here, we try to provide an intuitive explanation on the “magic”. Consider edge i and edge j where $\alpha_i > \alpha_j$ (recall that α_i is the position of edge i in the ground-truth edge sequence as defined in the main text). In the first step of our method, for any edge l with $\alpha_i > \alpha_l > \alpha_j$, the probability of correctly obtaining the reconstructed pairwise order of both edge pairs (i, l) and (j, l) is x^2 , which is the largest among all four possible combinations of pairwise orders since $x > 0.5$ (see Table R1). Therefore, when $\alpha_i - \alpha_j$ is not too small, by the law of large

numbers, we could generally obtain sufficient intermediate edges that have correct reconstructed pairwise orders with edge i and edge j , enabling the correct reconstructed ranking $\hat{\alpha}_i > \hat{\alpha}_j$ in the vote-based ranking algorithm of the second step. This can be more easily understood through an analogy with a biased dice. Imagine a four-sided dice, one of which has a higher probability than the other three sides. As long as you roll the dice for enough number of times, you are able to detect that the dice is biased.

Table R1. The ranking result of edge i and edge j based on the reconstructed pairwise order of edge pairs (i, l) and (j, l) for any edge l with $\alpha_i > \alpha_l > \alpha_j$.

Reconstructed order of (i, l)	Reconstructed order of (j, l)	Ranking result of edge i and edge j	Probability
$\hat{\alpha}_i > \hat{\alpha}_l$	$\hat{\alpha}_l > \hat{\alpha}_j$	$\hat{\alpha}_i > \hat{\alpha}_j$ (correct)	x^2
$\hat{\alpha}_i > \hat{\alpha}_l$	$\hat{\alpha}_l < \hat{\alpha}_j$	None	$x(1 - x)$
$\hat{\alpha}_i < \hat{\alpha}_l$	$\hat{\alpha}_l > \hat{\alpha}_j$	None	$x(1 - x)$
$\hat{\alpha}_i < \hat{\alpha}_l$	$\hat{\alpha}_l < \hat{\alpha}_j$	$\hat{\alpha}_i < \hat{\alpha}_j$ (incorrect)	$(1 - x)^2$

REVIEWERS' COMMENTS

Reviewer #2 (Remarks to the Author):

I am happy with the new version of the manuscript, so I recommend its publication.